# AGOF: A GFlowNet-Guided 2-Opt Framework for Vehicle Routing Problems

## Abstract

The 2-Opt algorithm is a widely used classical search method in vehicle routing problems (VRPs). However, existing learning-based approaches designed for 2-Opt rely on autoregressive (AR) architectures, which suffer from limited generalization and high computational overhead. In this work, we propose the first non-autoregressive (NAR) framework for 2-Opt, which addresses the generalization and efficiency limitations of prior AR-based models by reducing the complexity of the input space, enhancing the robustness of the reward landscape, and eliminating the need for repeated inference during optimization. To enable effective training within this framework, we introduce A GFlowNet-guided 2-Opt Framework (AGOF), which leverages the reward–edge alignment capabilities of Generative Flow Network (GFlowNet) to provide accurate and generalizable edge evaluations for guiding 2-Opt swaps. Furthermore, we design Exploration beyond Local Optima (ELO) to inject perturbations into the optimization process, helping the model escape local optimal solutions. Extensive experiments demonstrate that AGOF not only outperforms existing GFlowNet- and 2-Opt-based methods but also has favorable generalization and computation efficiency.

## 1 Introduction

Vehicle routing problems (VRPs) are a class of combinatorial optimization problems that aim to determine the most efficient routes for a fleet of vehicles to deliver goods or services to a set of customers. VRPs occur in a wide range of real-world applications, including operations management (Kim et al., 2025a; Konstantakopoulos et al., 2022; Feng & Ye, 2021), supply chain (Bai et al., 2022; Hasani Goodarzi & Zegordi, 2020; Ancele et al., 2021), and infrastructure planning (Pan et al., 2024; Li et al., 2018; Seo & Asakura, 2021), where optimized routing is crucial for minimizing operational costs and enhancing overall efficiency. In recent years, learn-to-search has emerged as one of the most promising paradigms for solving VRPs (Bengio et al., 2021b), motivating a series of studies that leverage machine learning to enhance search algorithm (Cooray & Rupasinghe, 2017; Cao et al., 2023; Chen et al., 2023; Sobhanan et al., 2025), among which the 2-Opt stands out as a prevalent local search strategy.

As a classical improvement operator, 2-Opt iteratively refines a given route by removing two non-adjacent edges and reconnecting the resulting segments in a reversed order, thereby reducing the total tour length. Specifically, it removes the edges $(v_{i-1}, v_i)$ and $(v_j, v_{j+1})$, and reconnects the route through $(v_{i-1}, v_j)$ and $(v_i, v_{j+1})$, while reversing the subpath between nodes $v_i$ and $v_j$. The performance of 2-Opt is highly sensitive to the choice of node pair $(i, j)$, as different selections can lead to dramatically different improvements. Traditionally, 2-Opt employs a greedy strategy that exhaustively evaluates all pairs and selects the one yielding the largest immediate cost reduction. Although simple and straightforward in practice, this myopic approach often traps the search in local optima and fails to explore more promising regions of the solution space.

In light of these shortcomings, Wu et al. (2021) explored integrating Transformer architectures with 2-Opt to guide edge selection. Building on this, Ma et al. (2021) proposed DACT, a model with network architectures specifically designed to align with the 2-Opt procedure, yielding improved performance. Afterwards, Ma et al. (2023) introduced NeuOpt, which extends 2-Opt to more complex k-Opt variants, thereby expanding the search space. Although these models achieve strong empirical results, they share key limitations, i.e., most notably poor generalization and high compu-

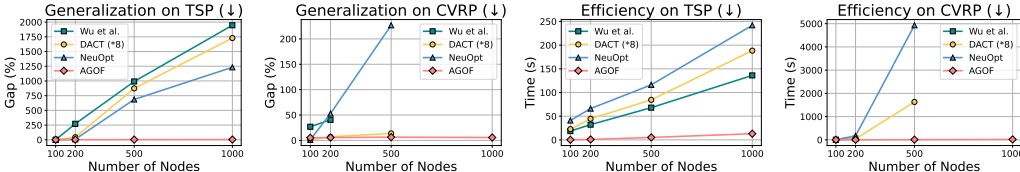

Figure 1: Generalization and Efficiency performance of different 2-Opt-based neural models on TSP and CVRP benchmarks, where all models are trained only on 100-node instances (lower gap and time indicates better performance). For CVRP, Wu et al. (2021) failed on 500- and 1000-node instances, DACT and NeuOpt failed on 1000-node instances, due to memory limitations.

tational overhead. A primary cause lies in their reliance on autoregressive (AR) architectures, where the model processes the entire route as input at each decision step. This design leads to a large and complex input space for two main reasons: (i) there is an exponential number of possible routes for a single instance, each of which can be an input at the decision step, and (ii) the relative ordering of nodes within each route must be explicitly encoded, as it significantly influences AR-based inference. Together, these factors inflate the input dimensionality, and create a reward landscape lack of robustness, which, as discussed in Sec. 3.1, hampers generalization to unseen instances, especially those of larger scale. Particularly, a comparison of the generalization performance of these models (against our AGOF) is presented in the first two plots in Figure 1. Moreover, the sequential nature of AR models requires repeatedly invoking the neural network at every decision step of swap process, leading to substantial computational overhead (against our AGOF) during inference, as exhibited in the last two plots in Figure 1.

To address these limitations, we propose a non-autoregressive (NAR) framework for 2-Opt that offers improved generalization ability and lower computational burden. First, the NAR framework takes only the graph as input, which reduces the complexity of the input space and eliminates the need for explicit positional features. As analyzed in Sec. 3.1 and the results shown in Figure. 1, this results in a more robust reward landscape and enhanced generalization performance. Moreover, the NAR framework requires only a single forward pass for inference, which significantly reduces computational overhead compared to autoregressive (AR) models. However, learning an effective NAR model for 2-Opt remains challenging, as it demands accurate evaluation over all edge pairs rather than optimizing a single solution trajectory. This requirement makes conventional reinforcement learning ill-suited, as it typically aims at reward maximization along a single trajectory. In routing problems, such a focus drives the model toward approximating the best route and often overlooks accurate estimation of many alternative routes under limited interactions. To address this, we adopt the Generative Flow Network (GFlowNet) (Bengio et al., 2021a), which instead learns to fit the entire reward distribution, enabling more accurate estimation of rewards across all possible edge combinations. Building on this insight, we introduce A GFlowNet-guided 2-Opt Framework (AGOF), which leverages the reward–edge alignment capabilities of GFlowNet to facilitate the training of the NAR model, and thus fully exploits its generalization capability and computational efficiency. To further enhance search effectiveness, we introduce an exploration mechanism named Exploration beyond Local Optima (ELO), which injects perturbations to help AGOF escape local minima. Our contributions are summarized as follows:

- We present the first non-autoregressive (NAR) framework for 2-Opt, which significantly improves generalization and notably reduces computational overhead through a smaller input space, a smoother reward landscape, and single-pass inference.

- We introduce a Generative Flow Network (GFlowNet)-guided 2-Opt framework (AGOF), which leverages the reward–edge alignment capability of GFlowNet to efficiently train the NAR model. To further improve search performance, we propose Exploration beyond Local Optima (ELO), which helps model to escape from local minima.

- We conduct extensive experiments on synthetic and real-world benchmarks (e.g., TSPLib and CVRPLib), demonstrating that AGOF consistently outperforms GFlowNet-based and 2-Opt-based baselines and generalizes well to varying scales and more complex instances.

**Note**. Our goal is not to surpass SOTA neural VRP solvers in general, but to advance the specific line of 2-Opt-based neural methods. By identifying their key limitations, we propose a strong alternative that substantially improves both the generalization and efficiency of neural 2-Opt solvers. **All code and data will be made publicly available**.

## 2  RELATED WORK

**Learning to search.** There has been growing interest in leveraging learning-based methods to enhance traditional search algorithms for combinatorial optimization problems (COPs) (Bengio et al., 2021b; Zhao et al., 2022; Yao et al., 2025; Cheng et al., 2022; Ye et al., 2023). A number of recent studies have particularly focused on applying such techniques to improve 2-Opt local search for solving VRPs. Wu et al. (2021) made an early attempt to directly learn 2-Opt by proposing a deep reinforcement learning framework based on a Transformer architecture, which autonomously learns policies for selecting position pairs in 2-Opt. Building on this foundation, Ma et al. (2021) proposed DACT, which replaces the vanilla attention mechanism with Dual-Aspect Collaborative Attention (DAC-Att) and introduces cyclic positional encoding to enhance structural awareness. Subsequently, NeuOpt (Ma et al., 2023) extended the traditional 2-Opt to k-Opt, which allows exploration of more broader neighborhoods and yields better solution quality. In addition, it leveraged a dynamic data augmentation to enhance search diversity. Although these works achieve favorable results, they all suffer from limited generalization and computational inefficiency due to their reliance on AR architectures. Different from them, we propose an NAR model, i.e., GFlowNet-guided 2-Opt (AGOF) that significantly enhances the generalization and efficiency.

**GFlowNet for solving combinatorial optimization.** Recent works have explored the application of Generative Flow Network (GFlowNet) to solve COPs. Zhang et al. (2023) first proposed using GFlowNet to solve classic COPs such as Maximum Independent Set, Maximum Clique, Minimum Dominating Set, and Max-Cut, demonstrating their effectiveness in learning diverse high-quality solutions. In the context of VRPs, GFACS (Chen et al., 2023) integrates GFlowNet with Ant Colony Optimization (ACO) to enhance solution quality by guiding the pheromone initialization. AGFN (Zhang et al., 2025) further advances this direction by using GFlowNet in an end-to-end manner to directly construct complete routes. While these methods leverage GFlowNet to guide or construct solutions, they primarily focus on route generation. In contrast, our work introduces a novel use of GFlowNet for route refinement via edge swap. Specifically, we employ GFlowNet to guide the 2-Opt swap process by learning reward-aligned edge evaluations.

## 3  METHODOLOGY

### 3.1  AR VS. NAR FOR 2-OPT IN VRP

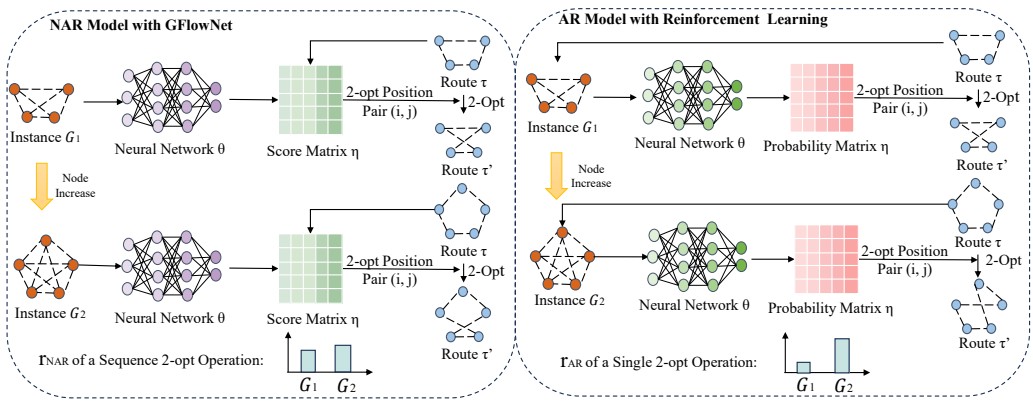

Figure 2: Comparison of AR and NAR models for 2-opt learning. **Left:** Non-autoregressive(NAR) model with GFlowNet uses only the instance $\mathcal{G}$ as input. A neural network with parameters $\theta$ predicts the score matrix $\eta$, which is repeatedly used to select a position pair $(i, j)$ for 2-opt to derive improved route $\tau'$. This one-shot inference mechanism avoids repeated forward passes, and results in significantly higher computational efficiency. **Right:** Autoregressive (AR) models such as Wu et al. (2021) and DACT (Ma et al., 2021) take both the instance $\mathcal{G}$ and the current route $\tau$ as input. They generate the probability matrix $\eta$ conditioned on $\tau$ via a neural network $\theta$. This path-dependent design introduces the lack of robustness in the reward landscape and requires inference at every step, significantly increasing computational cost.

We investigate the underlying causes of the differences in generalization ability and computational efficiency between autoregressive (AR) and non-autoregressive (NAR) frameworks in learning 2-Opt. An overview of these contributing factors is exhibited in Figure 2.

**Generalization analysis.** To better illustrate the generalization gap between AR and NAR frameworks in learning 2-Opt, we highlight two key factors that fundamentally differentiate their behaviors: the complexity of their input spaces and the robustness of the reward landscape. The robustness here refers to the property that when the input graph undergoes a small modification, the model's action(for AR) or heatmap(for NAR), scoring function and the resulting trajectory reward change only slightly and in a stable manner. In this context, robustness characterizes how stable the model's behavior remains under small perturbations to the input graph. Based on this definition, the following theoretical analysis aims to show that the reward of AR trajectories can vary significantly under such perturbations, whereas the reward of NAR trajectories exhibits much smaller change. We also provide additional empirical experiment in the Appendix 4.4 to demonstrate it.

**In AR-based frameworks**, the model takes as input a VRP instance $\mathcal{G} = (\mathcal{V}, \mathcal{E})$ along with the information of a complete route $\tau = (v_0, \ldots, v_n)$, including the predecessor and successor of each node, where $\mathcal{V}$ is the set of nodes, $\mathcal{E}$ is the set of edges, and each $v_i$ denotes a node in the instance. The model then outputs a probability matrix $\eta$ explicitly conditioned on the input path $\tau$, where each entry $\eta_{i,j}$ denotes the likelihood of applying a 2-Opt move between positions $i$ and $j$, with the most probable pair selected for the swap. This mapping can be expressed as:

$$f_\theta^{\mathrm{AR}} : (\mathcal{G}, \tau) \to (i^*, j^*) \quad \text{where} \quad (i^*, j^*) = \arg\max_{(i,j)} \eta_{i,j}. \tag{1}$$

Consequently, the input space $\mathcal{X}$ contains all feasible path:

$$\mathcal{X}_{\mathrm{AR}} = \mathcal{T}(\mathcal{G}), \tag{2}$$

where $|\mathcal{T}(\mathcal{G})| \sim n!$ for each instance $\mathcal{G}$. As a result, the model must generalize across an exponentially large space of highly variable routes. In addition, since the model is based on 2-Opt, it also requires encoding the detailed relative position information of each node within the route, including features such as the spatial coordinates of its predecessor and successor nodes. This further contributes to the overall complexity of the input.

Moreover, the output space induced by $\tau$ lacks inherent structure or robustness. Even when two input paths $\tau_1$ and $\tau_2$ differ only slightly (e.g., by swapping two nodes or exhibiting minor variations in node count), their corresponding 2-Opt predictions can differ significantly. This discrepancy arises because even minor modifications in the route can substantially change the relative position feature of nodes that are specifically required for AR 2-Opt model, leading to distinct input features and, consequently, different decisions by the model. Formally, this lack of robustness is characterized by:

$$\|\tau_1 - \tau_2\| \ll \epsilon \not\Rightarrow \|f_\theta^{\mathrm{AR}}(\tau_1) - f_\theta^{\mathrm{AR}}(\tau_2)\| \ll \epsilon. \tag{3}$$

And in the AR setting, the model only observes the outcome of its own prediction and lacks access to preceding or subsequent swap. As a result, the reward signal is defined by the cost reduction induced by a single 2-Opt operation:

$$r_{\mathrm{AR}}(\tau) \propto \mathrm{cost}(\tau) - \mathrm{cost}(\mathrm{Opt2}(\tau, f_\theta^{\mathrm{AR}}(\tau))), \tag{4}$$

where $\mathrm{Opt2}(\tau, f_\theta^{\mathrm{AR}}(\tau_1))$ is the path obtained by applying a 2-Opt move on $\tau$ and $\mathrm{cost}(\tau)$ is defined as:

$$\mathrm{cost}(\tau) = \sum_{k=1}^{n} d(v_{k-1}, v_k), \tag{5}$$

with $d(v_{k-1}, v_k)$ representing the distance between consecutive nodes in the path. Moreover, the reward signal $r_{\mathrm{AR}}$ is highly sensitive to small changes in the scale of input graph $\mathcal{G}$. This is because even a slight increase in the scale of $\mathcal{G}$ may introduce minor variations in the node count of $\tau$, which can lead to significant differences in 2-Opt predictions $(i, j) = f_\theta^{\mathrm{AR}}(\tau)$ as described in Eq. (3), resulting in large reward fluctuations. Formally,

$$\|\mathcal{G}_1 - \mathcal{G}_2\| \ll \epsilon \not\Rightarrow \|r_{\mathrm{AR}}(\tau_1, f_\theta^{\mathrm{AR}}(\tau_1)) - r_{\mathrm{AR}}(\tau_2, f_\theta^{\mathrm{AR}}(\tau_2))\| \ll \epsilon. \tag{6}$$

This lack of robustness in the reward landscape encourages overfitting to the path-specific patterns and impairs generalization ability across instances.

**While for NAR framework**, it directly conditions solely on the VRP instance $\mathcal{G} = (\mathcal{V}, \mathcal{E})$ rather than on the specific path, and learns a mapping:

$$f_\theta^{\text{NAR}} : \mathcal{G} \to \eta, \tag{7}$$

where $\eta \in \mathbb{R}^{n \times n}$ is a score matrix that guides the 2-Opt move between each position pair $(i, j)$ for all routes through a series of comparative computations. This design shifts the focus from memorizing path-level patterns to learning structure properties of instances that govern good 2-Opt regions across many possible solutions. Specifically, the input spaces $\mathcal{X}_{\text{NAR}}$ and output spaces $\mathcal{Y}_{\text{NAR}}$ are given by:

$$\mathcal{X}_{\text{NAR}} = \{\mathcal{G}\}, \quad \mathcal{Y}_{\text{NAR}} = \mathbb{R}_{\geq 0}^{n \times n}, \tag{8}$$

where $\mathcal{X}_{\text{NAR}}$ contains $\mathcal{G}$ as input, which represents a significant reduction in input space compared to the factorial $\mathcal{X}_{\text{AR}} = \mathcal{T}(\mathcal{G})$ in the AR setting. This simplification reduces the learning complexity and enhances model's generalization ability across instances.

Meanwhile, $\mathcal{Y}_{\text{NAR}}$ defines a structured and stable scoring matrix over all 2-Opt action pairs, providing a robust and structured output space in contract with AR models. This robustness stems from the structural stability of the input graph $\mathcal{G}$ and the stability of the score matrix $\eta$, which is generated by a continuous neural network. Specifically, the output $\eta$ remains stable as the scale of $\mathcal{G}$ increases, without significant changes in permutation. This is because the NAR model operates directly on the graph structure rather than a solution trajectory, and its input representation is typically based on node features and pairwise relations which is inherently robust encountering changes in graph size and invariant to node permutations. As a result, adding more nodes to $\mathcal{G}$ does not fundamentally alter the local feature patterns, allowing the model to produce stable and generalizable score predictions. Therefore, unlike AR models, the NAR mapping is robust:

$$\|\mathcal{G}_1 - \mathcal{G}_2\| \ll \epsilon \Rightarrow \|f_\theta^{\text{NAR}}(\mathcal{G}_1) - f_\theta^{\text{NAR}}(\mathcal{G}_2)\| \ll \epsilon. \tag{9}$$

Finally, instead of relying on a single-step reward from a specific 2-Opt move in AR, NAR framework defines a global reward based on the total cost of the final path obtained after applying a sequence of 2-Opt moves to a given input path $\tau$. This design avoids the use of local, stepwise rewards, and the immediate gain from a single 2-Opt move does not necessarily reflect its long-term contribution to the final solution quality. In contrast, a global reward provides a more holistic learning signal that captures the cumulative effect of an entire sequence of decisions:

$$r_{\text{NAR}}(\tau') \propto \text{cost}\left(\underbrace{\text{Opt2} \circ \cdots \circ \text{Opt2}}_{K \text{ times}}(\tau, \eta)\right), \tag{10}$$

where $\tau'$ denotes the final path obtained after applying $K$ successive 2-Opt moves guided by the score matrix $\eta$. Since $\eta = f_\theta^{\text{NAR}}(\mathcal{G})$ is a robust function of the input graph $\mathcal{G}$, small variations in the scale of $\mathcal{G}$ lead to small changes in $\eta$ as described in Eq. (9). As the final path $\tau'$ is generated through a deterministic sequence of 2-Opt moves conditioned on $\eta$, these changes result in only slight variations in $\tau'$. Given that the cost function is stable over path permutations, the overall reward defined as the cost of the final path, is therefore a robust function of $\mathcal{G}$.

$$\|\mathcal{G}_1 - \mathcal{G}_2\| \ll \epsilon \Rightarrow \|r_{\text{NAR}}(\tau'_1) - r_{\text{NAR}}(\tau'_2)\| \ll \epsilon. \tag{11}$$

This reward robustness promotes robust learning and better generalization across VRP instances.

**Computational efficiency analysis.** Another important distinction between AR and NAR frameworks lies in their computational efficiency. **AR models** generate solutions step by step, with each decision based on the previously constructed solution. This decoding process involves repeated inference steps, which result in high computational overhead. In contrast, **NAR models** generate the entire decision structure in a single forward pass without relying on intermediate solutions. This one-shot inference mechanism eliminates the need for step-by-step decoding, and substantially reduces inference time.

## 3.2 GFLOWNET-GUIDED NAR FOR 2-OPT

Learning an effective NAR model for 2-Opt is particularly challenging. Unlike constructive models, which only need to maximize reward for a single route and accurately evaluate a limited set of

related edges, 2-Opt must reliably assess all possible edge pairs, as any of them may be selected and compared during the swapping process. This limitation renders traditional reinforcement learning paradigm widely used in VRP domain (Kwon et al., 2020; Ye et al., 2023), which focuses on deriving single best trajectories and evaluates only related edges precisely, fundamentally ill-suited for NAR-based 2-Opt that demands comprehensive edge assessment.

To address this limitation, we adopt Generative Flow Network (GFlowNet) (Bengio et al., 2021a) as a more suitable learning approach for NAR-based 2-Opt. Rather than converging to one best trajectory, GFlowNet aims to learn the entire reward distribution, enabling more accurate estimation of rewards across all possible edge combinations, thereby achieving effective reward–edge alignment. The training objective of GFlowNet is to ensure the sampling distribution satisfies:

$$P_F(x) \propto R(x). \tag{12}$$

In VRP scenarios, $x$ denotes a complete trajectory that derived after a sequence of 2-Opt moves, and $R(x)$ is its associated reward, defined by the quality of the final solution. To achieve this objective, we adopt the Trajectory Balance (TB) objective (Malkin et al., 2022), a widely used training strategy for GFlowNet. Unlike previous AR works (Wu et al., 2021; Ma et al., 2021) that learn from individual steps, TB enables the model to learn from entire trajectories, allowing rewards to be assigned to each edge along the path. The TB loss is defined as:

$$\mathcal{L}_{\text{TB}}(\tau, \theta) = \left( \log \frac{Z_\theta(\mathcal{G}) \cdot \prod_{t=0}^{T-1} P_F^\theta(v_{t+1} \mid v_{\leq t})}{R(\tau) \cdot \prod_{t=0}^{T-1} P_B^\theta(v_t \mid v_{\geq t+1})} \right)^2, \tag{13}$$

where $\theta$ denotes the parameters of a graph neural network (GNN) and $Z_\theta(\mathcal{G})$ represents source flow, which are described in Appendix A.1 and Appendix A.2, respectively. The reward of $\tau$, $R(\tau)$, measures the advancement of the solution over other routes, computed as:

$$-\log R(\tau) = \text{cost}(\tau) - \frac{1}{T} \sum_{t=1}^{T} \text{cost}(\tau_t), \tag{14}$$

where $\text{cost}(\tau)$ is defined in Eq. (5), and $\{\tau_1, \tau_2, \cdots, \tau_T\}$ are routes obtained from the same instance during training. $P_B^\theta$ in Eq. (13) are backward probabilities, which are determined by the graph structure and the route $\tau$. $P_F^\theta$ in Eq. (13) denotes the forward transition probabilities in the GFlowNet. Here, $P_F^\theta$ is computed from the learned score matrix $\eta \in \mathbb{R}^{n \times n}$, where each entry $\eta_{ij}$ represents the model's predicted contribution of edge $e_{ij}$ to the overall trajectory reward:

$$P_F^\theta(v_{t+1} \mid v_{\leq t}) = \frac{\eta_{v_t, v_{t+1}}}{\sum_{e_{a,b} \in \mathcal{E}} \eta_{a,b}}. \tag{15}$$

The composite gain value $\Delta(i, j)$ using the entries of $\eta$ associated with the involved edges, is computed as follows to determine which position pair $(i, j)$ is selected for 2-Opt:

$$\Delta(i, j) = \eta_{i-1, j} + \eta_{i, j+1} - \eta_{i-1, i} - \eta_{j, j+1}, \tag{16}$$

where $\Delta(i, j)$ represents the predicted improvement in solution quality resulting from applying the 2-Opt move between positions $i$ and $j$. At each step $t$, we compute $\Delta(i, j)$ for all valid 2-Opt candidates, forming a matrix $\Delta(\tau, \eta)$ that captures the potential improvement of each possible swap given the route $\tau$ and score matrix $\eta$. The pair $(i^*, j^*)$ with the maximum gain is selected:

$$(i^*, j^*) = \arg\max \Delta(\tau, \eta). \tag{17}$$

This selection strategy aligns with the GFlowNet objective $P_F(x) \propto R(x)$ in Eq. (12), which encourages assigning higher forward probabilities to trajectories with greater rewards. By choosing the edge pair that maximizes $\Delta(i, j)$ and thus contributes most to increasing $P_F$, the model effectively guides the search toward better solutions. Compared to traditional 2-Opt, this approach leverages learned global structure to make more informed swap decisions, leading to faster convergence and better solution quality.

Table 1: Performance comparison between AGOF and baseline methods on the TSP and CVRP.

| Task | I | Method | 100 | | | 200 | | | 500 | | | 1000 | | |
|---|---|---|---|---|---|---|---|---|---|---|---|---|---|---|
| | | | Obj.↓ | Gap(%)↓ | Time(s)↓ | Obj.↓ | Gap(%)↓ | Time(s)↓ | Obj.↓ | Gap(%)↓ | Time(s)↓ | Obj.↓ | Gap(%)↓ | Time(s)↓ |
| TSP | | LKH | 7.75 | – | 2.01 | 10.72 | – | 9.88 | 16.55 | – | 32.49 | 23.14 | – | 115.35 |
| | | GFACS | 8.78 | 12.85 | 0.59 | 13.02 | 21.46 | 1.83 | 22.86 | 38.13 | 9.82 | 41.60 | 79.78 | 21.69 |
| | | AGFN | 8.49 | 9.55 | 0.06 | 11.85 | 10.54 | 0.10 | 19.08 | 15.28 | 0.28 | 27.15 | 17.32 | 0.78 |
| | | POMO(*8) | 7.77 | 0.26 | 0.14 | 10.90 | 1.68 | 0.28 | 20.15 | 21.75 | 0.67 | 32.74 | 41.49 | 3.15 |
| | | LCH-Regret | 7.76 | 0.13 | 0.30 | 10.94 | 2.05 | 0.56 | 20.66 | 24.84 | 2.17 | 30.65 | 32.45 | 5.68 |
| | | GNARKD | 7.83 | 1.03 | 0.07 | 13.10 | 22.20 | 0.12 | 25.26 | 52.53 | 0.35 | 37.24 | 60.93 | 0.81 |
| | | MDAM | 8.38 | 7.02 | 2.79 | 14.05 | 31.06 | 5.97 | 25.31 | 52.93 | 8.79 | 39.16 | 69.23 | 19.02 |
| | 3000 | Wu et al. | 7.91 | 2.06 | 6.13 | 41.38 | 286.01 | 10.05 | 182.83 | 1004.71 | 16.31 | 480.03 | 1974.46 | 45.57 |
| | | DACT(*8) | 7.81 | 0.77 | 6.84 | 17.61 | 64.27 | 13.59 | 173.80 | 950.15 | 24.75 | 448.96 | 1840.19 | 66.48 |
| | | NeuOpt | **7.76** | **0.13** | 9.42 | 11.90 | 11.01 | 21.70 | 132.53 | 700.79 | 37.96 | 313.02 | 1252.72 | 74.99 |
| | | AGOF | 7.87 | 1.55 | **0.08** | **10.99** | **2.52** | **0.30** | **17.25** | **4.23** | **1.51** | **24.43** | **5.57** | **5.03** |
| | 5000 | Wu et al. | 7.86 | 1.42 | 10.71 | 40.46 | 277.43 | 18.50 | 181.81 | 693.72 | 30.41 | 474.66 | 1951.25 | 61.25 |
| | | DACT(*8) | 7.79 | 0.52 | 11.80 | 16.70 | 55.78 | 22.15 | 166.04 | 903.27 | 43.93 | 438.88 | 1796.63 | 96.17 |
| | | NeuOpt | **7.75** | **0.03** | 15.70 | 11.77 | 9.79 | 36.17 | 131.20 | 692.75 | 64.02 | 310.58 | 1242.18 | 132.13 |
| | | AGOF | 7.85 | 1.29 | **0.11** | **10.96** | **2.24** | **0.54** | **17.21** | **3.98** | **2.04** | **24.24** | **4.74** | **6.77** |
| | 10000 | Wu et al. | 7.86 | 1.42 | 18.63 | 39.85 | 271.74 | 32.17 | 181.08 | 994.14 | 68.04 | 473.42 | 1945.89 | 136.09 |
| | | DACT(*8) | 7.76 | 0.13 | 22.77 | 15.40 | 43.66 | 44.57 | 155.21 | 837.82 | 84.50 | 423.24 | 1729.04 | 188.39 |
| | | NeuOpt | **7.75** | **0.03** | 40.95 | 11.67 | 8.86 | 66.60 | 130.08 | 685.98 | 116.26 | 308.21 | 1231.94 | 241.94 |
| | | AGOF | 7.83 | 1.03 | **0.17** | **10.90** | **1.96** | **0.94** | **17.14** | **3.56** | **5.15** | **24.23** | **4.71** | **13.01** |
| CVRP | | LKH | 15.57 | – | 60.25 | 28.04 | – | 157.94 | 63.32 | – | 834.80 | 120.53 | – | 4951.92 |
| | | GFACS | 19.26 | 23.70 | 1.96 | 34.55 | 23.22 | 4.18 | 78.34 | 23.72 | 12.93 | 150.47 | 24.84 | 26.10 |
| | | AGFN | 17.78 | 14.19 | 0.07 | 31.26 | 11.48 | 0.15 | 71.05 | 12.21 | 0.38 | 133.96 | 11.14 | 0.70 |
| | | POMO(*8) | 15.75 | 1.17 | 0.08 | 29.20 | 4.14 | 0.30 | 77.20 | 21.92 | 0.92 | 188.74 | 56.59 | 3.83 |
| | | LCH-Regret | 15.72 | 9.63 | 0.35 | 30.60 | 9.13 | 0.68 | 123.06 | 94.35 | 2.18 | 372.17 | 208.78 | 36.79 |
| | | GNARKD | 16.83 | 8.09 | 0.09 | 31.72 | 13.12 | 0.18 | 76.66 | 21.07 | 0.43 | 157.25 | 30.47 | 00.92 |
| | 3000 | Wu et al. | 21.20 | 36.16 | 1.31 | 41.47 | 47.90 | 16.75 | – | – | – | – | – | – |
| | | DACT(*8) | 16.86 | 8.29 | 1.17 | 30.92 | 10.27 | 14.98 | 72.81 | 14.99 | 521.1 | – | – | – |
| | | NeuOpt | 16.80 | 7.90 | 2.09 | 44.88 | 60.06 | 52.04 | 207.55 | 2277.80 | 1271.28 | – | – | – |
| | | AGOF | **16.53** | **6.17** | **0.43** | **29.86** | **6.49** | **0.70** | **68.05** | **7.47** | **2.44** | **131.70** | **9.27** | **5.49** |
| | 5000 | Wu et al. | 20.28 | 30.25 | 2.70 | 40.08 | 42.94 | 25.01 | – | – | – | – | – | – |
| | | DACT(*8) | 15.81 | 4.54 | 1.55 | 30.40 | 8.42 | 22.16 | 72.45 | 14.42 | 868.13 | – | – | – |
| | | NeuOpt | **15.79** | **1.41** | 4.94 | 43.58 | 55.42 | 83.03 | 207.24 | 272.29 | 2501.82 | – | – | – |
| | | AGOF | 16.46 | 5.72 | **0.75** | **29.72** | **5.99** | **1.17** | **67.77** | **7.03** | **4.31** | **127.40** | **5.70** | **9.56** |
| | 10000 | Wu et al. | 19.75 | 26.85 | 4.88 | 39.41 | 40.51 | 51.57 | – | – | – | – | – | – |
| | | DACT(*8) | 15.76 | 1.22 | 3.39 | 29.98 | 6.92 | 39.42 | 72.05 | 13.79 | 1631.73 | – | – | – |
| | | NeuOpt | **15.74** | **1.10** | 9.03 | 42.88 | 52.92 | 168.51 | 206.95 | 226.83 | 4930.95 | – | – | – |
| | | AGOF | 16.37 | 5.14 | **1.32** | **29.58** | **5.49** | **1.93** | **67.31** | **6.30** | **8.27** | **127.26** | **5.58** | **18.27** |

## 3.3 EXPLORATION BEYOND LOCAL OPTIMA

While performing 2-Opt optimization, it is common for the model to quickly converge to a local optimum from which no further improvement can be found using standard swap operations. Traditional approaches typically address this issue by restarting the search from a randomly initialized route and reapplying 2-Opt, but this strategy can be computationally inefficient and redundant.

To address this limitation, we propose an exploration mechanism, Exploration beyond Local Optima (ELO), which is triggered when the search process becomes trapped in a local optimum. Instead of discarding the current solution, our approach temporarily replaces the learned score matrix $\eta$ with the original distance matrix to introduce mild perturbations. These perturbations allow the search to gently deviate from the local optimum and continue exploring the solution space in a controlled manner. Although the perturbation relies on the traditional distance matrix used in classical 2-Opt methods, the primary contributor to performance improvement is still the learned score matrix $\eta$, as the perturbation is intentionally subtle. We further demonstrate the effectiveness of this design and our overall model through ablation studies in Sec. 4.2.

As shown in Algorithm 1 in Appendix, the procedure begins with a randomly initialized solution $\tau_0$ (Line 2) and iteratively applies 2-Opt moves guided by the learned score matrix $\eta$ (Lines 4–6) until a local optimum is reached. To escape from local optima, we introduce an exploration mechanism (ELO), which temporarily replaces $\eta$ with the original distance matrix $D$ to perform mild perturbations (Lines 11–14). The distance matrix $D$ is computed based on Euclidean distances between nodes. After perturbation, $\eta$-guided local search is resumed (Lines 15–18). If the perturbed solution improves upon the best solution found so far, the perturbation level is reset; otherwise, it increases gradually (Line 19). Once the perturbation limit is exceeded, a random restart is triggered (Line 22). This procedure continues until the maximum number of iterations is reached.

## 4 EXPERIMENT

### 4.1 COMPARISON STUDIES

**Baselines.** To evaluate the effectiveness of AGOF, we compare it against a diverse set of baseline methods, including: the classical heuristic solver LKH (Helsgaun, 2017) and the neural

Table 2: Ablation study on the effectiveness of each component in AGOF. The values reported in the table represent gap (%) relative to LKH, with lower values indicating better performance.

| I | Method | TSP | | | | CVRP | | | |
|---|--------|-----|-----|-----|------|------|-----|-----|------|
| | | 100 | 200 | 500 | 1000 | 100 | 200 | 500 | 1000 |
| 3000 | Conventional 2-opt | 2.97 | 5.88 | 8.82 | 10.03 | 9.18 | 10.02 | 9.82 | 10.80 |
| | AGOF w/o ELO | 2.19 | 4.01 | 5.86 | 6.66 | 8.86 | 8.80 | 8.77 | 9.27 |
| | AGOF w/ Ran-ELO | 1.81 | 3.36 | 5.20 | 5.70 | 7.65 | 7.31 | 7.68 | 9.27 |
| | AGOF w/ dis-ELO | **1.55** | **2.52** | **4.23** | **5.57** | **6.17** | **6.49** | **7.47** | **9.27** |
| 5000 | Conventional 2-opt | 2.58 | 5.41 | 8.28 | 9.85 | 8.54 | 9.77 | 9.60 | 10.24 |
| | AGOF w/o ELO | 1.94 | 3.54 | 5.50 | 6.35 | 8.29 | 8.45 | 8.29 | 7.69 |
| | AGOF w/ Ran-ELO | 1.42 | 2.99 | 4.89 | 6.70 | 7.13 | 6.99 | 7.34 | 6.21 |
| | AGOF w/ dis-ELO | **1.29** | **2.24** | **3.98** | **4.74** | **5.72** | **5.99** | **7.03** | **5.70** |
| 10000 | Conventional 2-opt | 2.19 | 5.03 | 7.98 | 9.29 | 7.84 | 9.13 | 9.33 | 9.96 |
| | AGOF w/o ELO | 1.55 | 3.17 | 5.20 | 5.92 | 7.58 | 7.92 | 7.66 | 7.49 |
| | AGOF w/ Ran-ELO | 1.16 | 2.80 | 4.53 | 5.23 | 6.55 | 6.31 | 6.66 | 6.16 |
| | AGOF w/ dis-ELO | **1.03** | **1.96** | **3.56** | **4.71** | **5.14** | **5.49** | **6.30** | **5.58** |

*"w/" and "w/o" denote "with" and "without", respectively. "Dis-ELO" and "Ran-ELO" refer to distance-based and random-based perturbation strategies within the ELO module.

model POMO (Kwon et al., 2020), LCH-Regret (Sun et al., 2024), GNARKD (Xiao et al., 2024), MDAM (Xin et al., 2021); GFlowNet-based approaches, including GFACS (Kim et al., 2025b) and AGFN (Zhang et al., 2025); and Opt-based methods such as Wu et al. (2021), DACT (Ma et al., 2021), and NeuOpt (Ma et al., 2023). We conduct experiments on both TSP and CVRP benchmarks, and report the results in Table 1, where $I$ denotes the number of 2-Opt iterations. More experiment settings are presented in Appendix B. We also provide results for AGOF without ELO and compare them with existing 2-opt-based methods in the Appendix E.

**TSP.** Our model consistently delivers strong performance and robust generalization across all instance sizes. While the performance on 100- and 200-node instances is slightly below that of POMO and the 100-node result is slightly behind LCH-Regret, AGOF substantially outperforms POMO, LCH-Regret, GNARKD, and MDAM as the problem size increases, achieving increasingly larger margins on the larger instances. Compared to GFlowNet-based models, our approach (with $I = 3000$) consistently achieves better results than GFACS across all problem sizes while achieving significantly shorter runtime. For fairness, we disable auxiliary modules in GFACS that are unrelated to its core ACO mechanism. AGOF also surpasses AGFN in terms of solution quality. When compared to Opt-based methods, although our model does not always outperform all of them on 100-node instances, AGOF demonstrates greater advantages on larger-scale instances. Specifically, it achieves superior performance on 200-, 500-, and 1000-node instances with substantially lower runtime. This efficiency is largely attributed to our model's NAR framework, which significantly reduces computational burden.

**CVRP.** On the CVRP benchmark, AGOF consistently achieves remarkable performance, reflecting both high solution quality and effective generalization in more complex, constraint-rich routing scenarios. Similar to the TSP case, AGOF performs comparably to POMO on 100- and 200-node instances and LCH-Regret on 100-node instances, and exhibits markedly improved performance with POMO, LCH-Regret and GNARKD on other scale problems. Compared to GFlowNet-based models, AGOF yields lower objective values. Against Opt-based baseline methods such as DACT, Wu et al., and NeuOpt, our model consistently delivers competitive results. While AGOF may not always surpass these methods on 100-node instances, it shows strong overall performance. Moreover, the improvements in gap over Opt-based baselines on large-scale instances (such as 200-, 500-, 1000-node) underline AGOF's generalization ability. Notably, DACT, Wu et al., and NeuOpt fail to run on large instances due to out-of-memory errors, whereas AGOF maintains robust performance, demonstrating its superior computational efficiency.

**Larger instances.** Additional results on further larger instances (3000- and 5000-node TSP and CVRP) are reported in Table 6 and Appendix C for completeness.

## 4.2 ABLATION STUDIES

To validate the effectiveness of our model, we conduct a series of ablation studies. We first compare AGOF (without the proposed exploration mechanism, ELO) against the conventional distance-based

Table 3: Comparison with distributional RL and bayesian RL on TSP and CVRP.

| Task | I | Method | Obj.(100) | Gap(%) | Obj.(200) | Gap(%) | Obj.(500) | Gap(%) | Obj.(1000) | Gap(%) |
|------|---|--------|-----------|--------|-----------|--------|-----------|--------|------------|--------|
| TSP | 3000 | Distributional RL | 7.97 | 2.84 | 11.26 | 5.04 | 17.78 | 7.43 | 24.98 | 7.95 |
| | | Bayesian RL | 8.01 | 3.35 | 11.36 | 5.97 | 17.80 | 7.55 | 25.14 | 8.64 |
| | | GFlowNet | **7.87** | **0.08** | **10.99** | **0.30** | **17.25** | **1.51** | **24.43** | **5.03** |
| | 5000 | Distributional RL | 7.91 | 2.06 | 11.22 | 4.66 | 17.69 | 6.89 | 24.94 | 7.78 |
| | | Bayesian RL | 7.94 | 2.45 | 11.29 | 5.32 | 17.74 | 7.19 | 25.03 | 8.17 |
| | | GFlowNet | **7.85** | **1.29** | **10.96** | **2.24** | **17.21** | **3.98** | **24.24** | **4.74** |
| | 10000 | Distributional RL | 7.91 | 2.06 | 11.21 | 4.57 | 17.67 | 6.77 | 24.89 | 7.56 |
| | | Bayesian RL | 7.94 | 2.45 | 11.25 | 4.94 | 17.71 | 7.09 | 24.96 | 7.87 |
| | | GFlowNet | **7.83** | **1.03** | **10.90** | **1.96** | **17.14** | **3.56** | **24.23** | **4.71** |
| CVRP | 3000 | Distributional RL | 16.87 | 8.35 | 30.65 | 9.31 | 69.77 | 10.19 | 143.21 | 17.99 |
| | | Bayesian RL | 17.02 | 9.51 | 30.93 | 10.31 | 71.77 | 13.35 | 146.15 | 21.26 |
| | | GFlowNet | **16.53** | **6.17** | **29.86** | **6.49** | **68.05** | **7.47** | **131.70** | **9.27** |
| | 5000 | Distributional RL | 16.78 | 7.77 | 30.47 | 8.67 | 69.29 | 9.43 | 134.29 | 11.42 |
| | | Bayesian RL | 16.91 | 8.61 | 30.89 | 10.16 | 69.82 | 10.27 | 137.09 | 13.74 |
| | | GFlowNet | **16.46** | **5.72** | **29.72** | **5.99** | **67.77** | **7.03** | **127.40** | **5.70** |
| | 10000 | Distributional RL | 16.73 | 7.45 | 30.38 | 8.35 | 69.24 | 9.35 | 130.74 | 8.47 |
| | | Bayesian RL | 16.84 | 8.16 | 30.83 | 9.95 | 70.31 | 11.04 | 132.81 | 10.19 |
| | | GFlowNet | **16.37** | **5.14** | **29.58** | **5.49** | **67.31** | **6.30** | **127.26** | **5.58** |

2-Opt algorithm to demonstrate the effectiveness of our learned model. We then evaluate the impact of the ELO module by comparing AGOF with and without ELO, and further analyze different perturbation strategies by testing distance-based and random-based variants of ELO. All experiments are conducted on TSP and CVRP instances of size 100, 200, 500, and 1000, with 3000, 5000, and 10000 training iterations. Finally, to assess the benefit of using GFlowNet for training 2-Opt model, we compare it against a reinforcement learning (RL) baseline under the same training protocol. We also provide the different initialization method analysis in Appendix F

**Compassion with conventional 2-Opt.** We compare AGOF without ELO to the conventional 2-Opt algorithm to evaluate the effectiveness of the learned model, using identical settings. As shown in the first and second rows of Table 2, AGOF without ELO consistently outperforms the conventional 2-Opt on both TSP and CVRP across all instance sizes and iteration counts.

**Comparison of AGOF with and without ELO.** To evaluate the contribution of the ELO module, we compare AGOF with and without ELO. The results, shown in the second, third, and fourth rows of Table 2, indicate that incorporating ELO consistently improves performance across all problem sizes and iteration settings on both TSP and CVRP.

**Comparison of ELO variants.** We compare the performance of AGOF using random-based ELO (Ran-ELO) and distance-based ELO (Dis-ELO). The results are shown in the third and fourth rows of Table 2. In the Ran-ELO setting, the perturbation matrix is randomly generated, with entries sampled uniformly from the interval [0,1] and subsequently normalized. In contrast, Dis-ELO leverages the inverse of pairwise distances between nodes to guide perturbation. As shown in Table 2, Dis-ELO consistently outperforms Ran-ELO across all instance sizes and training iterations on both TSP and CVRP, highlighting the advantage of Dis-ELO.

**Comparison of distance based ELO's effect.** Although Dis-ELO uses a distance-based perturbation matrix, the performance gain primarily comes from the learning-based nature of AGOF. This is evidenced by the fact that the conventional distance-based 2-Opt algorithm (first row of Table 2) performs worse than the distance-based AGOF with Dis-ELO (fourth row of Table 2).

**Comparison of GFlowNet training and reinforcement learning.** Figure 3 in Appendix compares the training performance of GFlowNet and reinforcement learning (RL) method which is widely used in VRP domain (Kwon et al., 2020; Ye et al., 2023), on TSP and CVRP under identical experimental settings. Following the same training protocol, GFlowNet consistently achieves lower objective values across both tasks. This performance gain can be attributed to GFlowNet's inherent suitability for the 2-Opt optimization within the NAR framework. We additional provide the NAR model using distributional Reinforcement Learning (RL) framework (Bellemare et al., 2017) and Bayesian RL Ghavamzadeh et al. (2015) in Table 3, to show that NAR model can solve generalization problem in 2-opt, and GFlowNet is the best one to enhance NAR as it can better achieve edge-reward alignment.

Table 4: Generalization on TSPLib and CVRPLib. The values reported in the table represent the gap (%) relative to current best-known solutions, with lower values indicating better performance.

|  | Model/Node | AGOF | POMO | GFACS | AGFN | Wu et al. | DACT | NeuOpt |
|---|---|---|---|---|---|---|---|---|
| TSPLib | <200 | **0.44** | 4.70 | 29.60 | 19.41 | 18.68 | 1.46 | 0.52 |
|  | 200–500 | **9.25** | 20.55 | 28.57 | 24.63 | 234.08 | 1555.80 | 1568.63 |
|  | >500 | **5.29** | 41.55 | 35.09 | 31.07 | 257.75 | 3481.57 | 2736.49 |
| CVRPLib | <200 | 12.65 | 8.38 | 49.63 | 17.13 | 14.25 | 3.61 | **3.22** |
|  | 200–500 | **11.00** | 31.15 | 65.50 | 13.60 | – | 22.38 | 65.59 |
|  | >500 | **11.47** | 38.17 | 30.02 | 13.32 | – | – | – |

Table 5: Comparison of reward sensitivity to graph perturbation. '-' indicates Out-of-Memory issues for those neural baselines.

|  | TSP | | | | CVRP | | | |
|---|---|---|---|---|---|---|---|---|
|  | $\Delta$Obj.(100) | $\Delta$Obj.(200) | $\Delta$Obj.(500) | $\Delta$Obj.(1000) | $\Delta$Obj.(100) | $\Delta$Obj.(200) | $\Delta$Obj.(500) | $\Delta$Obj.(1000) |
| Wu et al.(3000) | 0.306 | 0.452 | 0.573 | 0.648 | 0.194 | 0.297 | - | - |
| DACT(*8)(3000) | 0.095 | 0.496 | 0.528 | 0.615 | 0.131 | 0.135 | 0.166 | - |
| NeuOpt(3000) | 0.038 | 0.359 | 0.376 | 0.391 | 0.176 | 0.508 | 0.870 | - |
| AGOF(3000) | **0.028** | **0.017** | **0.011** | **0.015** | **0.128** | **0.122** | **0.120** | **0.126** |

## 4.3 GENERALIZATION TO REAL-WORLD BENCHMARK DATASETS

We evaluate the generalization ability of AGOF on two widely-used real-world benchmarks: TSPLib (Reinelt, 1991) and CVRPLib (Uchoa et al., 2017). For TSPLib, we use the standard 50 test instances ranging in size from 50 to over 1000 nodes. For CVRPLib, we evaluate on 100 instances covering a variety of problem sizes, from small (<200 nodes) to large (>500 nodes). All neural models, including our AGOF and the 2-Opt-based baselines, DACT, Wu et al., and NeuOpt, run 10000 iterations for Opt. Performance is measured by the optimality gap relative to the known best solutions provided by the datasets. As shown in Table 4, our model (AGOF) achieves the best generalization performance on TSPLib across all instance sizes, consistently outperforming all baselines.

## 4.4 EMPIRICAL STUDY

We provide an empirical study to examine how small changes in the input graph influence the resulting reward, and to show that our method responds much smaller variations than other AR models. In this experiment, we introduce a minimal structural modification by adding a single new node to existing datasets of 100-, 200-, 500-, and 1000-nodes, and then measure how much the final reward changes after running each method. This setup directly evaluates the sensitivity of different models to small perturbations in the graph. The results, presented in Table 5, show that our model exhibits far smaller reward fluctuations across all instance sizes when compared with 2-opt based AR methods.

## 5 CONCLUSION

In this work, we propose AGOF, the first framework to successfully apply a non-autoregressive (NAR) architecture to 2-Opt for vehicle routing problems. By leveraging Generative Flow Network (GFlowNet) to provide reward–edge aligned training, AGOF fully exploits the generalization capability and computational efficiency of NAR inference. To further enhance search effectiveness, we introduce Exploration beyond Local Optima (ELO), which helps the model escape local minima. Extensive experiments on both synthetic datasets and real-world benchmarks (TSPLib and CVR-PLib) demonstrate that AGOF not only outperforms existing GFlowNet-based and 2-Opt-based neural methods but also exhibits strong generalization ability across problem scales and distributions. An exciting future direction is to extend AGOF beyond 2-Opt to more general k-Opt operations. While the primary focus of this work is to advance neural 2-Opt solvers, we will also include comparisons with SOTA neural VRP solvers beyond the GFlowNet- and Opt-based categories.

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

## A SUPPLEMENTARY METHODOLOGY

### A.1 GRAPH NEURAL NETWORK

Graph Neural Network (GNN) encodes raw node information $\mathcal{V}$ and edge distances $\mathcal{E}$ into initial features $\mathbf{h}_i^0$ and $\mathbf{e}_{ij}^0 \in \mathbb{R}^d$, and updates them as following:

$$\mathbf{h}_i^{l+1} = \mathbf{h}_i^l + \text{ACT}\left(\text{BN}\left(\mathbf{W}_1^l \mathbf{h}_i^l + \mathcal{A}\left(\sigma(\mathbf{e}_{ij}^l) \odot \mathbf{W}_2^l \mathbf{h}_j^l\right)\right)\right), \quad (18)$$

$$\mathbf{e}_{ij}^{l+1} = \mathbf{e}_{ij}^l + \text{ACT}\left(\text{BN}\left(\mathbf{W}_3^l \mathbf{e}_{ij}^l + \mathbf{W}_4^l \mathbf{h}_i^l + \mathbf{W}_5^l \mathbf{h}_j^l\right)\right), \quad (19)$$

where $\mathbf{W}_1^l$, $\mathbf{W}_2^l$, $\mathbf{W}_3^l$, $\mathbf{W}_4^l$, and $\mathbf{W}_5^l$ are learnable weight matrices at layer $l$; $\mathbf{h}_i^l$ and $\mathbf{e}_{ij}^l$ are node and edge features at layer $l$; $\sigma$ is the sigmoid function; $\odot$ is element-wise multiplication; $\mathcal{A}$ is mean pooling over neighbors; BN denotes batch normalization; and ACT is the activation function (e.g., SiLU (Elfwing et al., 2018)).

### A.2 SOURCE FLOW

The source flow $Z_\theta(\mathcal{G})$ serves as the initial flow value in GFlowNet, indicating the total probability mass emitted from the source state. And it is computed from the final-layer edge embeddings of the GNN $e^d$ as follows:

$$Z_\theta(\mathcal{G}) = \mathbf{W}_1 \cdot \text{ReLU}(\mathbf{W}_2 \cdot e^d + b_1) + b_2, \quad (20)$$

where $\mathbf{W}_1$, $\mathbf{W}_2$, $b_1$, and $b_1$ are learnable parameters, and ReLU is the activation function.

### A.3 ALGORITHM

---

**Algorithm 1** GFlowNet-guided 2-opt with exploration beyond local optimal

---

1: **Input:** random initial solution $\tau_0$, score matrix $\eta$, distance matrix $D$, max iterations $I$, max perturbation $Z$, initial perturbation $z_0$, increased perturbation $\xi$
2: $\tau \leftarrow \tau_0$, $\tau^* \leftarrow \tau_0$, $z \leftarrow z_0$, $s \leftarrow 0$
3: **while** $s < N$ **do**
4:     **while** $\max \Delta(\tau, \eta) > 0$ and $s < I$ **do**           ▷ Step 1: $\eta$-guided 2-opt
5:         $(i^*, j^*) \leftarrow \arg\max \Delta(\tau, \eta)$
6:         $\tau \leftarrow \text{2-opt}(\tau, i^*, j^*)$, $s \leftarrow s+1$
7:     **end while**
8:     $\tau^* \leftarrow \tau$ if $\text{cost}(\tau) < \text{cost}(\tau^*)$              ▷ update best solution
9:     **while** $z \leq Z$ and $s < I$ **do**
10:         $i \leftarrow 1$
11:         **while** $i < \min(z, I-s)$ and $\max \Delta(\tau, D) > 0$ **do**   ▷ Step 2: Exploration beyond local optima
12:             $(i^*, j^*) \leftarrow \arg\max \Delta(\tau, D)$
13:             $\tau \leftarrow \text{2-opt}(\tau, i^*, j^*)$, $i \leftarrow i + 1$, $s \leftarrow s+1$
14:         **end while**
15:         **while** $\max \Delta(i, j) > 0$ and $s < I$ **do**         ▷ Step 3: Resume $\eta$-guided 2-opt
16:             $(i^*, j^*) \leftarrow \arg\max \Delta(i, j)$
17:             $\tau \leftarrow \text{2-opt}(\tau, i^*, j^*)$, $s \leftarrow s+1$
18:         **end while**
19:         $z \leftarrow z_0$ if $\text{cost}(\tau) < \text{cost}(\tau^*)$, else $z \leftarrow z+\xi$     ▷ adjust perturbation level
20:         $\tau^* \leftarrow \tau$ if $\text{cost}(\tau) < \text{cost}(\tau^*)$
21:     **end while**
22:     $\tau \leftarrow$ random solution, $z \leftarrow z_0$            ▷ restart if perturbation failed
23: **end while**
24: **return** best solution $\tau^*$

---

# B  SUPPLEMENTARY EXPERIMENTS

**Hyperparameter.** The GNN dimension is set to 32, and the network depth is set to 12. We use a batch size of 5 and run the training for 1000 steps. Training on 100-node TSP instances takes approximately 20 minutes, while training on 100-node CVRP instances takes about 2 hours. All experiments are run for 5 times to test the model's stability. During training, AGOF optimizes 10 routes per instance to encourage route diversity, whereas only one route is optimized during testing. The perturbation mechanism is governed by three hyperparameters: the initial perturbation strength $z_0 = 20$, the perturbation increment $\xi = 20$, and a maximum perturbation threshold $Z = 100$. The hyperparameter settings for all baseline methods follow their respective original papers.

**Dataset and Experimental Setting.** Following prior works (Kwon et al., 2020; Wu et al., 2021; Ma et al., 2021; Kim et al., 2025b; Zhang et al., 2025), we conduct experiments on both the TSP and CVRP. Training and test instances are generated using a uniform distribution, as in (Kwon et al., 2020). For our AGOF, all initial routes are randomly generated. All experiments are conducted on a machine equipped with an NVIDIA RTX 3090 GPU and an AMD EPYC 7702P CPU. To ensure fairness, all models used for comparison are trained on 100-node instances. The evaluation includes 10,000 randomly sampled 100-node instances and additional test sets consisting of 128 instances for each of the 200-, 500-, and 1000-node cases. More experiment details are presented in Appendix.

**Random Seed Initialization.**

To ensure robustness and reproducibility, we employed five fixed seed values, i.e., 0, 10, 100, 1000, and 10000, consistently across all components, including data loading, model initialization, and random initialization.

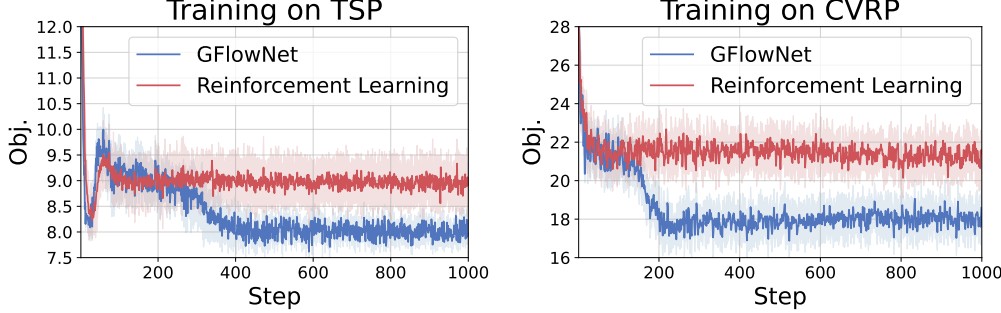

Figure 3: Comparison of training performance between RL and GFlowNet on TSP and CVRP. GFlowNet demonstrates better final performance in both tasks.

# C  EVALUATION ON FURTHER LARGER INSTANCES.

Table 6: Comparison gap (%) on TSP (left) and CVRP (right).

| Nodes | TSP | | | | | | | CVRP | | | | | | |
|---|---|---|---|---|---|---|---|---|---|---|---|---|---|---|
| | AGOF | POMO | GFACS | AGFN | Wu et al. | DACT | NeuOpt | AGOF | POMO | GFACS | AGFN | Wu et al. | DACT | NeuOpt |
| 3000 | **4.89** | 62.25 | 70.84 | 37.41 | – | 3525.16 | 3062.73 | **-4.58** | 84.15 | 3.38 | -3.00 | – | – | – |
| 5000 | **5.27** | 71.39 | *86.26 | 58.04 | – | – | – | **-6.47** | 111.79 | 1.24 | -5.26 | – | – | – |

*The gap is measured against LKH (10000) for TSP and LKH (1000) for CVRP on those further larger instances, as running 10,000 iterations on CVRP would be prohibitively time-consuming. '-' indicates Out-of-Memory issues for those neural baselines.

To evaluate the scalability of our method on further larger instances, we report results on both 3000- and 5000-node problems for TSP and CVRP, as shown in Table 6. All Opt-based neural models, including AGOF, Wu et al., DACT, and NeuOpt, run with 20,000 iterations to ensure high-quality reference solutions. Across both TSP and CVRP, our AGOF achieves superior performance

Table 7: Comparison with LKH on CVRP.

| | Obj.(100) | Gap(%) | Time(s) | Obj.(200) | Gap(%) | Time(s) | Obj.(500) | Gap(%) | Time(s) | Obj.(1000) | Gap(%) | Time(s) |
|---|---|---|---|---|---|---|---|---|---|---|---|---|
| LKH | 15.85 | 1.80 | 0.50 | 29.65 | 5.74 | 0.67 | 69.60 | 10.27 | 2.75 | 134.74 | 11.79 | 5.88 |
| AGOF(3000) | 16.53 | 6.17 | 0.43 | 29.86 | 6.49 | 0.70 | 68.05 | 7.47 | 2.44 | 131.70 | 9.27 | 5.49 |
| LKH | 15.80 | 0.32 | 0.75 | 29.04 | 3.57 | 1.22 | 68.72 | 8.53 | 4.32 | 131.52 | 9.12 | 9.75 |
| AGOF(5000) | 16.46 | 5.72 | 0.75 | 29.72 | 5.99 | 1.17 | 67.77 | 7.03 | 4.31 | 127.40 | 5.70 | 9.56 |
| LKH | 15.78 | 0.19 | 1.40 | 28.75 | 2.53 | 1.99 | 67.39 | 6.43 | 8.52 | 127.61 | 5.87 | 18.67 |
| AGOF(10000) | 16.37 | 5.14 | 1.32 | 29.58 | 5.49 | 1.93 | 67.31 | 6.30 | 8.27 | 127.26 | 5.58 | 18.27 |

Table 8: Comparison with Gurobi.

| | Obj.(200)↓ | | | Obj.(500)↓ | | | Obj.(1000)↓ | | |
|---|---|---|---|---|---|---|---|---|---|
| | Obj. | Gap(%) | Time(s) | Obj. | Gap(%) | Time(s) | Obj. | Gap(%) | Time(s) |
| Gurobi | 29.91 | 6.68 | 600 | 73.18 | 15.52 | 600 | – | – | 1200 |
| AGOF(10000) | **29.58** | **5.49** | **1.93** | **67.31** | **6.30** | **8.27** | **127.26** | **5.58** | **18.27** |

with consistently low optimality gaps, significantly outperforming all learning-based baselines and even other optimization-based methods. Notably, several methods including Wu et al., DACT, and NeuOpt failed to complete on certain problem sizes due to out-of-memory (OOM) issues, highlighting the computational challenges posed by these large-scale benchmarks. In contrast, our AGOF maintains robust performance and runtime stability across all instance sizes, further demonstrating its scalability and efficiency.

# D    COMPARISON WITH LKH AND GUROBI

Although our work primarily focuses on addressing the generalization limitations of 2-opt–based models, we also provide comparisons with LKH and Gurobi (Gurobi Optimization, LLC, 2024) on CVRP in Table 7 and Table 8 for completeness. The results show that AGOF outperforms LKH on the 500-node and 1000-node instances, and surpasses Gurobi on the 200-, 500-, and 1000-node instances.

# E    COMPARISON WITH BASELINES AND AGOF WITHOUT ELO

We include results for AGOF without ELO and compare them with existing 2-opt-based methods. As shown in Table 9, AGOF without ELO can still outperform other baselines on 200, 500, 1000. These results show that the improvement of AGOF arises from both the design of GFlowNet and the incorporation of ELO.

# F    DIFFERENT INITIALIZATION METHOD

To study the sensitivity of different methods to the quality of the initial solution, we conducted additional experiments using greedy initialization and report the results in Table 10. The results show that providing a higher-quality initial solution does not substantially change the overall conclusions. For our method, the performance with greedy initialization is almost the same across all instance sizes compared with using a random initial solution, confirming that AGOF is largely insensitive to the initial tour quality. For other 2-opt-based baselines, greedy initialization leads to a slight improvement on the 200-, 500-, and 1000-node instances compared to initial greedy results. However, even with this advantage, their performance remains worse than AGOF. On the 100-node instances, their performance with greedy initialization is similar to that with random initialization.

Table 9: Comparison of Gap(%) with baselines and AGOF without ELO on TSP and CVRP. '-' indicates Out-of-Memory issues for those neural baselines.

| Task | Node Size | 3000 | | | | 5000 | | | | 10000 | | | |
|---|---|---|---|---|---|---|---|---|---|---|---|---|---|
| | | Wu et al. | DACT(*8) | NeuOpt | AGOF | Wu et al. | DACT(*8) | NeuOpt | AGOF | Wu et al. | DACT(*8) | NeuOpt | AGOF |
| TSP | 100 | 2.06 | 0.77 | **0.13** | 2.97 | 1.42 | 0.52 | **0.03** | 2.58 | 1.42 | 0.13 | **0.03** | 2.19 |
| | 200 | 286.01 | 64.27 | 11.01 | **5.88** | 277.43 | 55.78 | 9.79 | **5.41** | 271.74 | 43.66 | 8.86 | **5.03** |
| | 500 | 1004.71 | 950.15 | 700.79 | **8.82** | 693.72 | 903.27 | 692.75 | **8.28** | 994.14 | 837.82 | 385.98 | **7.98** |
| | 1000 | 1974.46 | 1840.19 | 1252.72 | **10.03** | 1951.25 | 1796.63 | 1242.18 | **9.85** | 1945.89 | 1729.04 | 1231.94 | **9.29** |
| CVRP | 100 | 36.16 | 8.29 | **7.90** | 9.18 | 30.25 | 4.54 | **1.41** | 8.54 | 26.85 | 1.22 | **1.10** | 5.14 |
| | 200 | 47.90 | 10.27 | 60.06 | **10.08** | 42.94 | **8.42** | 55.42 | 9.77 | 40.51 | 6.92 | 52.92 | **5.49** |
| | 500 | - | 14.99 | 2277.80 | **9.82** | - | 14.42 | 272.29 | **9.60** | - | 13.79 | 226.83 | **6.30** |
| | 1000 | - | - | - | **10.80** | - | - | - | **10.24** | - | - | - | **5.58** |

Table 10: Comparison with greedy initialization on TSP

| Task | I | Method | Obj.(100) | Gap(%) | Obj.(200) | Gap(%) | Obj.(500) | Gap(%) | Obj.(1000) | Gap(%) |
|---|---|---|---|---|---|---|---|---|---|---|
| TSP | | Traditional Greedy | 9.13 | 17.81 | 12.85 | 19.87 | 20.07 | 21.27 | 28.80 | 24.46 |
| | 3000 | Wu et al. | 7.90 | 1.94 | 12.84 | 19.78 | 20.64 | 24.71 | 28.70 | 24.03 |
| | | DACT(*8) | **7.80** | **0.64** | 12.56 | 17.16 | 20.51 | 23.93 | 28.04 | 21.13 |
| | | NeuOpt | 7.83 | 1.03 | 11.84 | 10.45 | 20.84 | 25.92 | 28.96 | 25.15 |
| | | AGOF | 7.85 | 12.90 | **10.95** | **2.15** | **17.23** | **4.11** | **24.36** | **5.27** |
| | 5000 | Wu et al. | 7.84 | 1.61 | 12.46 | 16.23 | 20.57 | 24.29 | 28.63 | 23.73 |
| | | DACT(*8) | **7.78** | **0.38** | 12.32 | 14.93 | 20.34 | 22.90 | 27.83 | 20.28 |
| | | NeuOpt | 7.79 | 0.52 | 11.74 | 9.51 | 20.80 | 25.68 | 28.88 | 24.81 |
| | | AGOF | 7.82 | 0.90 | **10.93** | **1.96** | **17.20** | **3.93** | **24.22** | **4.67** |
| | 10000 | Wu et al. | 7.84 | 1.16 | 12.27 | 14.46 | 20.52 | 23.99 | 28.61 | 23.64 |
| | | DACT(*8) | **7.77** | **0.26** | 12.18 | 13.62 | 20.30 | 22.66 | 27.79 | 20.10 |
| | | NeuOpt | 7.78 | 0.39 | 11.71 | 9.24 | 20.78 | 25.56 | 28.84 | 24.63 |
| | | AGOF | 7.82 | 0.90 | **10.90** | **1.68** | **17.15** | **3.63** | **24.20** | **4.58** |
| CVRP | | Traditional Greedy | 20.01 | 28.52 | 35.07 | 25.07 | 76.80 | 21.29 | 148.63 | 23.31 |
| | 3000 | Wu et al. | 19.34 | 24.21 | 30.54 | 8.92 | - | - | - | - |
| | | DACT(*8) | 16.68 | 7.13 | 30.66 | 9.34 | 72.73 | 14.86 | - | - |
| | | NeuOpt | 16.60 | 6.62 | 34.36 | 22.54 | 75.02 | 18.48 | - | - |
| | | AGOF | **16.47** | **5.78** | **29.73** | **6.03** | **67.85** | **7.15** | 130.89 | 8.60 |
| | 5000 | Wu et al. | 19.24 | 23.57 | 30.18 | 7.63 | - | - | - | - |
| | | DACT(*8) | 15.82 | 1.61 | 30.42 | 8.49 | 72.17 | 13.98 | - | - |
| | | NeuOpt | **15.77** | **1.25** | 34.25 | 22.15 | 74.82 | 18.16 | - | - |
| | | AGOF | 16.42 | 5.46 | **29.60** | **5.53** | **67.43** | **6.49** | 127.30 | 5.62 |
| | 10000 | Wu et al. | 19.20 | 23.31 | 29.78 | 6.21 | - | - | - | - |
| | | DACT(*8) | 15.75 | 1.56 | 30.03 | 7.10 | 72.72 | 14.85 | - | - |
| | | NeuOpt | **15.75** | **1.15** | 34.10 | 21.61 | 74.70 | 17.97 | - | - |
| | | AGOF | 16.38 | 5.20 | **29.56** | **5.42** | **67.26** | 62.22 | 127.30 | 5.60 |

These results support our claim that AGOF does not rely on the initial solution and is robust to its quality, while other 2-opt-based methods benefit slightly from a better initialization but still do not reach the performance of AGOF.

## G STATEMENT ON USAGE OF LARGE LANGUAGE MODELS

During the preparation of this manuscript, large language models (LLMs) were engaged only for stylistic editing and refinement of the written text. Their involvement was confined to enhancing readability and fluency, without influencing the study's design, methodology, data analysis, or scientific conclusions. All intellectual content and final interpretations remain solely the responsibility of the authors.

