# OpenReview forum: "AGOF: A GFlowNet-Guided 2-Opt Framework for Vehicle Routing Problems"
_ICLR.cc/2026/Conference — Submitted to ICLR 2026_

### Official Review · Reviewer_pYHa · 2025-10-19

**Soundness:** 2
**Presentation:** 3
**Contribution:** 3
**Rating:** 6
**Confidence:** 3

**Summary:**

This paper proposes a non-autoregressive (NAR) algorithm for solving Vehicle Routing Problems (VRPs) named AGOF by leveraging the 2-Opt heuristic. It utilizes GFlowNet to guide the NAR model in performing 2-Opt operations and incorporates an exploration mechanism to escape local optima. AGOF is designed to overcome the limitations of autoregressive (AR) models in terms of generalization ability and computational efficiency.

To address the traditional Reinforcement Learning (RL) paradigm's focus on maximizing rewards for single trajectories, the authors introduce GFlowNet to learn the entire reward distribution. This is achieved by training with the Trajectory Balance objective function, thereby enabling reward-edge alignment. The score matrix generated by GFlowNet is then used to calculate the potential improvement for each 2-Opt. Furthermore, an Exploration beyond Local Optima (ELO) mechanism is introduced to tackle the inherent tendency of 2-Opt to quickly converge to local optima.

Experimental results demonstrate that AGOF exhibits remarkably fast inference speed and strong generalization capabilities, showing very small optimality gaps on large-scale problems. Ablation studies further confirm the individual contributions of both modules to the overall performance.

**Strengths:**

1.This paper is the first to propose the application of the Non-Auto Regressive model to the 2-Opt method. This framework breaks through the dependence of the 2-Opt method on the AR framework, thereby being able to overcome the inherent difficulties of the AR framework.
2.this paper thoroughly compares the differences between the AR algorithm and the NAR algorithm in dealing with the 2-Opt problem, making it easier for readers to understand.
3.The experimental results show that the algorithm proposed in this paper can significantly reduce the inference time. It can solve large-scale problems in TSP and CVRP that were previously unsolvable by the 2-Opt algorithm, and surpass the computational efficiency and effectiveness of the previous 2-Opt-based algorithms.

**Weaknesses:**

1.Although the author claimed that their goal is not to surpass SOTA neural VRP solver in general but to advance the specific line of 2-Opt-based neural methods, not comparing the algorithm with the well-known general neural VRP solver would still undermine its soundness to some extent.
2.This article does not provide any code files, making it impossible to verify the reproducibility and fairness of the comparison. Please specify which programming language was used to accomplish this work. If possible, please submit your code files. As far as I know, executing the same code in C++ is significantly faster than in Python. If the author does not clarify this, it will greatly undermine the credibility of the claim regarding the reduction in runtime, as there is no guarantee that the comparison experiment was conducted fairly. If the authors' method is implemented in a more efficient language (e.g., C++) while the baselines are implemented in Python, the reported speedup could be largely attributed to this implementation bias rather than the algorithm itself.
3. Please see Questions to find more details. Explaining them will make this paper more clear.
Some typos:
1.Line625. ‘max isterationps’ ==> ‘max iterations’

**Questions:**

1.As Demonstrated in weaknesses, which programming language was mainly used to accomplish this work?
2.Could you explain why you the distance matrix D as an alternative to the scoring matrix in ELO?
3.Could you explain why you choose GFlowNet to overcome the limitation of AR, not other methods like distributional RL or Bayesian RL?

---

> ### Author Response · Authors · 2025-11-27
>
> We sincerely thank the reviewer for these encouraging and constructive comments. We appreciate your attention of our contributions, including the introduction of a non-autoregressive framework for 2-Opt, the detailed comparison between AR and NAR formulations, and the demonstrated improvements in inference speed and scalability. Your feedback is greatly appreciated. Our responses for your suggestions and concerns are provided below.
>
> ### **W1. Comparison with Other Algorithm**
>
> We thank the reviewer for raising this important point. Although our primary goal is to advance the line of 2-opt based neural methods rather than to surpass general purpose neural VRP solvers, we agree that including such comparisons can enhance the completeness and soundness of the study. Therefore, we include additional comparisons with representative neural VRP algorithms, and the results are provided in the revised paper (Section 4.1 and Table 1). These results show that our model is able to surpass these neural solvers.
>
> **Table D. Comparison with More Baselines on TSP and CVRP**
>
> |                    | Obj.↓(100) | Gap(%)↓  | Time(s) | Obj.↓(200) | Gap(%)↓  | Time(s) | Obj.↓(500) | Gap(%)↓  | Time(s) | Obj.↓(1000) | Gap(%)↓  | Time(s) |
> | ------------------ | ---------- | -------- | ------- | ---------- | -------- | ------- | ---------- | -------- | ------- | ----------- | -------- | ------- |
> | GNARKD(TSP)[1]     | 7.83       | 1.03     | 0.07    | 13.10      | 22.20    | 0.12    | 25.26      | 52.53    | 0.35    | 37.24       | 60.93    | 0.81    |
> | LCH-Regret(TSP)[2] | **7.76**   | **0.13** | 0.30    | 10.94      | 2.05     | 0.56    | 20.66      | 24.84    | 2.17    | 30.65       | 32.45    | 5.68    |
> | MDAM(TSP)[3]       | 8.38       | 7.02     | 2.79    | 14.05      | 31.06    | 5.97    | 25.31      | 52.93    | 8.79    | 39.16       | 69.23    | 19.02   |
> | AGOF(TSP)          | 7.83       | 1.03     | 0.17    | **10.90**  | **1.96** | 0.94    | **17.14**  | **3.56** | 5.15    | **24.23**   | **4.71** | 13.01   |
> | GNARKD(CVRP)       | **16.83**  | **8.09** | 0.09    | 31.72      | 13.12    | 0.18    | 76.66      | 21.07    | 0.43    | 157.25      | 30.47    | 0.92    |
> | LCH-Regret(CVRP)   | 15.72      | 9.63     | 0.35    | 30.60      | 9.13     | 0.68    | 123.06     | 94.35    | 2.18    | 372.17      | 208.78   | 36.79   |
> | AGOF(CVRP)         | 16.37      | 5.14     | 1.32    | **29.58**  | **5.49** | 1.93    | **67.31**  | **6.30** | 8.27    | **127.26**  | **5.58** | 18.27   |
>
> ### **W2, Q1. Programming Language**
>
> We thank the reviewer for raising this important concern regarding reproducibility and implementation fairness. All experiments in our paper, including our method and all baselines, are implemented entirely in Python to ensure a fair and consistent comparison environment. No components of our method are implemented in C++ or other compiled languages. To further support reproducibility, we have publicly released the source code, and the code repository is available at: **https://anonymous.4open.science/r/AGOF-9FC8**
>
> ### **W3, Q2. Distance Matrix as ELO**
>
> We thank the reviewer for the question. Our choice of using the distance matrix D as an alternative scoring matrix in ELO is empirically supported by our results. As shown in Section 4.2 and Table 2 in the paper, incorporating D into the ELO mechanism consistently yields better solution quality than using the other scoring matrix. Our analysis suggests that the distance matrix provides a mild yet effective perturbation to the current local optimum: it enables the model to escape suboptimal regions while avoiding excessive disturbance that would significantly degrade the solution quality.
>
> ### **W3, Q3. GFlowNet Compare to Distributional RL or Bayesian RL**
>
>  The motivation for adopting a NAR framework is that NAR models naturally alleviate the generalization limitations observed in 2-opt based AR methods, especially when scaling to larger or unseen instances. Within this NAR setting, we choose GFlowNet because its formulation is particularly suitable for addressing the edge–reward alignment issue, which is a core challenge in NAR-based 2-opt model.
>
> To further clarify this design choice, we additionally provide NAR variants using distributional reinforcement learning and Bayesian reinforcement learning in Table J and Table K. These results show that while NAR models generally improve the generalization ability of 2-opt, the GFlowNet-enhanced NAR model performs the best, as it achieves more consistent alignment between edge evaluation and final reward. These new results and discussions have been incorporated into the revised paper in the Section 4.2 and Table 3.

---

> > ### Author Response · Authors · 2025-11-27
> >
> > **Table J. Comparison with Distributional RL and Bayesian RL on TSP**
> >
> > |                          | Obj.↓(100) | Gap(%)↓  | Obj.↓(200) | Gap(%)↓  | Obj.↓(500) | Gap(%)↓  | Obj.↓(1000) | Gap(%)↓  |
> > | ------------------------ | ---------- | -------- | ---------- | -------- | ---------- | -------- | ----------- | -------- |
> > | Distributional RL(3000)  | 7.97       | 2.84     | 11.26      | 5.04     | 17.78      | 7.43     | 24.98       | 7.95     |
> > | Bayesian RL(3000)        | 8.01       | 3.35     | 11.36      | 5.97     | 17.80      | 7.55     | 25.14       | 8.64     |
> > | GFlowNet(3000)           | **7.87**   | **0.08** | **10.99**  | **0.30** | **17.25**  | **1.51** | **24.43**   | **5.03** |
> > | Distributional RL(5000)  | 7.91       | 2.06     | 11.22      | 4.66     | 17.69      | 6.89     | 24.94       | 7.78     |
> > | Bayesian RL(5000)        | 7.94       | 2.45     | 11.29      | 5.32     | 17.74      | 7.19     | 25.03       | 8.17     |
> > | GFlowNet(5000)           | **7.85**   | **1.29** | **10.96**  | **2.24** | **17.21**  | **3.98** | **24.24**   | **4.74** |
> > | Distributional RL(10000) | 7.91       | 2.06     | 11.21      | 4.57     | 17.67      | 6.77     | 24.89       | 7.56     |
> > | Bayesian RL(10000)       | 7.94       | 2.45     | 11.25      | 4.94     | 17.71      | 7.09     | 24.96       | 7.87     |
> > | GFlowNet(10000)          | **7.83**   | **1.03** | **10.90**  | **1.96** | **17.14**  | **3.56** | **24.23**   | **4.71** |
> >
> > **Table J. Comparison with Distributional RL and Bayesian RL on CVRP**
> >
> > |                          | Obj.↓(100) | Gap(%)↓  | Obj.↓(200) | Gap(%)↓  | Obj.↓(500) | Gap(%)↓  | Obj.↓(1000) | Gap(%)↓  |
> > | ------------------------ | ---------- | -------- | ---------- | -------- | ---------- | -------- | ----------- | -------- |
> > | Distributional RL(3000)  | 16.87      | 8.35     | 30.65      | 9.31     | 69.77      | 10.19    | 143.21      | 17.99    |
> > | Bayesian RL(3000)        | 17.02      | 9.51     | 30.93      | 10.31    | 71.77      | 13.35    | 146.15      | 21.26    |
> > | GFlowNet(3000)           | **16.53**  | **6.17** | **29.86**  | **6.49** | **68.05**  | **7.47** | **131.70**  | **9.27** |
> > | Distributional RL(5000)  | 16.78      | 7.77     | 30.47      | 8.67     | 69.29      | 9.43     | 134.29      | 11.42    |
> > | Bayesian RL(5000)        | 16.91      | 8.61     | 30.89      | 10.16    | 69.82      | 10.27    | 137.09      | 13.74    |
> > | GFlowNet(5000)           | **16.46**  | **5.72** | **29.72**  | **5.99** | **67.77**  | **7.03** | **127.40**  | **5.70** |
> > | Distributional RL(10000) | 16.73      | 7.45     | 30.38      | 8.35     | 69.24      | 9.35     | 130.74      | 8.47     |
> > | Bayesian RL(10000)       | 16.84      | 8.16     | 30.83      | 9.95     | 70.31      | 11.04    | 132.81      | 10.19    |
> > | GFlowNet(10000)          | **16.37**  | **5.14** | **29.58**  | **5.49** | **67.31**  | **6.30** | **127.26**  | **5.58** |
> >
> > **Typos**
> >
> > We thank the reviewer for pointing out the typo. We have corrected it in the revised version.
> >
> > [1]Distilling autoregressive models to obtain high-performance non-autoregressive solvers for vehicle routing problems with faster inference speed. AAAI.
> >
> > [2]Learning Encodings for Constructive Neural Combinatorial Optimization Needs to Regret. AAAI.
> >
> > [3]Multi-Decoder Attention Model with Embedding Glimpse for Solving Vehicle Routing Problems. AAAI.
> >
> > ------
> > We greatly appreciate the reviewer’s thoughtful feedback and suggestions, which have been very helpful for strengthening our work. If you have any additional questions or further recommendations, please feel free to let us know, and we would be glad to respond.

---

### Official Review · Reviewer_bCtj · 2025-10-24

**Soundness:** 3
**Presentation:** 3
**Contribution:** 2
**Rating:** 4
**Confidence:** 3

**Summary:**

The paper proposes AGOF, a novel non-autoregressive (NAR) framework for guiding 2-Opt local search in VRPs.
By leveraging GFlowNets for reward-aligned edge evaluations and introducing an Exploration beyond Local Optima (ELO) mechanism to escape local minima, AGOF addresses key limitations of prior AR models, including poor generalization to larger instances and high computational overhead.
Empirical results on synthetic and real-world benchmarks demonstrate superior solution quality, generalization, and efficiency.

**Strengths:**

- Introduces NAR approach for 2-Opt, creatively integrating GFlowNet for comprehensive edge-pair evaluation rather than single-trajectory maximization.
- The ELO mechanism is a simple yet effective addition for perturbation-based exploration, enhancing practical utility.
- The paper is well-organized, with intuitive explanations and thorough motivation for design choices.

**Weaknesses:**

- There is no comparison with solvers like Gurobi and other NAR methods[1] applicable to VRP problems.
- On small and medium-scale VRPs (<500), the solution quality of the proposed method did not show considerable advantages over baselines, which may limit its applicability.
- Section 3.1 mentioned that in AR framework, small differences on G corresponds small difference on r. It would be better if there is an experiments to demonstrate it.


[1] Xiao, Yubin, et al. "Distilling autoregressive models to obtain high-performance non-autoregressive solvers for vehicle routing problems with faster inference speed." Proceedings of the AAAI conference on artificial intelligence. Vol. 38. No. 18. 2024.

**Questions:**

see weaknesses.

- Both AR and NAR framework use the sampled trajectories, why the proposed NAR approach can achieve better quality results?

---

> ### Author Response · Authors · 2025-11-27
>
> We sincerely thank the reviewer for the positive and encouraging comments. We appreciate your appreciation of our non-autoregressive 2-opt design, the integration of GFlowNet for edge-pair evaluation, the effectiveness of the ELO mechanism, and the overall clarity and organization of the paper. Your feedback is highly motivating for us. Below, we provide our detailed responses with additional experiments to your comments and suggestions.
>
> ### **W1. Comparison with Gurobi and NAR**
>
> We thank the reviewer for the suggestion. Due to the very large scale of the 10,000-instance dataset for size 100, we were not able to run Gurobi on all of these cases due to time limit. As a result, we report Gurobi’s performance on 128 instances for sizes 200 and 500 with a 10 minute time limit each, and on 128 instances for size 1000 with a 60 minute time limit. The results in the Table I show that AGOF surpasses Gurobi on the 200- and 500- node instances, and the Gurobi cannot derives feasible solution in 60 minutes, which show AGOF's signficance and efficiency. Also, our model consistently surpasses GNARKD across the evaluated settings, which further validates the effectiveness of our design. These results, along with additional discussion, have been incorporated into the revised paper in Section 4.1 and Table 1.
>
> **Table I. Comparison with GNARKD, Using Gurobi as the Benchmark**
>
> |              | Obj.↓(100) | Gap(%)↓  | Obj.↓(200) | Gap(%)↓  | Obj.↓(500) | Gap(%)↓  | Obj.↓(1000) | Gap(%)↓  |
> | ------------ | ---------- | -------- | ---------- | -------- | ---------- | -------- | ----------- | -------- |
> | GNARKD(TSP)  | **7.83**   | 0.91     | 13.10      | 22.20    | 25.26      | 52.26    | 37.24       | 60.93    |
> | AGOF(TSP)    | **7.83**   | **0.90** | **10.90**  | **1.96** | **17.14**  | **3.32** | **24.23**   | **4.71** |
> | Gurobi(CVRP) | - | -| 29.91      |6.68 |  73.18      |15.52 |-| -|
> | GNARKD(CVRP) | 16.83      | 8.09     | 31.72      | 13.12    | 76.66      | 21.07    | 157.25      | 30.47    |
> | AGOF(CVRP)   | **16.37**  | **5.14** | **29.58**  | **5.49** | **67.31**  | **6.30** | **127.26**  | **5.58** |
>
> ### **W2. Solution Quality**
>
> Our work is motivated by the fundamental bottleneck that 2-opt based methods often struggle with generalization. As shown in our experiments, the proposed approach achieves better performance than existing 2-opt baselines at n=200, 500, and 1000, demonstrating stable generalization across different instance scales. Although the performance on very small instances is slightly behind several baselines developed in other modeling paradigms, the method surpasses them once the instance size reaches 500 and above. This trend suggests that the generalization improvements introduced by our approach become increasingly beneficial as the problem complexity grows. Overall, these results indicate that learning enhanced 2-opt models hold strong potential for medium and large VRP scenarios, where scalability and cross-distribution robustness are essential for practical deployment.
>
> ### **W3. Empirical Experiment**
>
> We thank the reviewer for this helpful suggestion. Following your advice, we conducted an additional experiment to examine how small changes in G affect the reward r. Specifically, we added a single node to the existing datasets at sizes 100, 200, 500, and 1000 and measured the resulting change in the final reward. As shown in Table C, our model exhibits significantly smaller reward fluctuations than other 2-opt based AR methods, indicating a more stable mapping from graph perturbations to reward outcomes. This further supports our claim in Section 3.1 that the proposed NAR formulation is less sensitive to minor structural changes in the input graph. These results have been incorporated into the revised paper in Section 4.4 and Table 5.

---

> ### Author Response · Authors · 2025-11-27
>
> **Table C. Comparison of Reward Sensitivity to Graph Perturbation**
>
> |                 | ΔObj.↓(100)(TSP) | ΔObj.↓(200)(TSP) | ΔObj.↓(500)(TSP) | ΔObj.↓(1000)(TSP) | ΔObj.↓(100)(CVRP) | ΔObj.↓(200)(CVRP) | ΔObj.↓(500)(CVRP) | ΔObj.↓(1000)(CVRP) |
> | --------------- | ---------------- | ---------------- | ---------------- | ----------------- | ----------------- | ----------------- | ----------------- | ------------------ |
> | Wu et al.(3000) | 0.306            | 0.452            | 0.573            | 0.648             | 0.194             | 0.297             | -                 | -                  |
> | DACT(*8)(3000)  | 0.095            | 0.496            | 0.528            | 0.615             | 0.131             | 0.135             | 0.166             | -                  |
> | NeuOpt(3000)    | 0.038            | 0.359            | 0.376            | 0.391             | 0.176             | 0.508             | 0.870             | -                  |
> | AGOF(3000)      | **0.028**        | **0.017**        | **0.011**        | **0.015**         | **0.128**         | **0.122**         | **0.120**         | **0.126**          |
>
> ### **Q1. Why NAR is Better**
>
> Although both AR and NAR frameworks rely on sampled trajectories, the key difference lies in their generalization behavior. Our work is motivated by the well-known generalization weakness of 2-opt based models, which often fail to transfer effectively to larger or unseen instances. The proposed NAR approach is specifically designed to overcome this bottleneck. In addition, the use of GFlowNet further helps the NAR framework address the edge reward alignment problem inherent in NAR models, enabling more consistent correspondence between edge evaluation and the final reward. These improvements collectively allow our method to surpass existing NAR approaches. We additionally provide the NAR model using distributional Reinforcement Learning (RL) framework and Bayesian RL in Table J and Table K, to show that NAR model can solve generalization problem in 2-opt, and GFlowNet is the strongest one to enhance NAR as it can better achieve edge-reward alignment and mitigate collapsing to a certain mode due to its diversity-seeking ability. We also incorporate these results in the Section 4.2 and Table 3 in the paper.
>
> **Table J. Comparison with Distributional RL and Bayesian RL on TSP**
>
> |                          | Obj.↓(100) | Gap(%)↓  | Obj.↓(200) | Gap(%)↓  | Obj.↓(500) | Gap(%)↓  | Obj.↓(1000) | Gap(%)↓  |
> | ------------------------ | ---------- | -------- | ---------- | -------- | ---------- | -------- | ----------- | -------- |
> | Distributional RL(3000)  | 7.97       | 2.84     | 11.26      | 5.04     | 17.78      | 7.43     | 24.98       | 7.95     |
> | Bayesian RL(3000)        | 8.01       | 3.35     | 11.36      | 5.97     | 17.80      | 7.55     | 25.14       | 8.64     |
> | GFlowNet(3000)           | **7.87**   | **0.08** | **10.99**  | **0.30** | **17.25**  | **1.51** | **24.43**   | **5.03** |
> | Distributional RL(5000)  | 7.91       | 2.06     | 11.22      | 4.66     | 17.69      | 6.89     | 24.94       | 7.78     |
> | Bayesian RL(5000)        | 7.94       | 2.45     | 11.29      | 5.32     | 17.74      | 7.19     | 25.03       | 8.17     |
> | GFlowNet(5000)           | **7.85**   | **1.29** | **10.96**  | **2.24** | **17.21**  | **3.98** | **24.24**   | **4.74** |
> | Distributional RL(10000) | 7.91       | 2.06     | 11.21      | 4.57     | 17.67      | 6.77     | 24.89       | 7.56     |
> | Bayesian RL(10000)       | 7.94       | 2.45     | 11.25      | 4.94     | 17.71      | 7.09     | 24.96       | 7.87     |
> | GFlowNet(10000)          | **7.83**   | **1.03** | **10.90**  | **1.96** | **17.14**  | **3.56** | **24.23**   | **4.71** |

---

> > ### Author Response · Authors · 2025-11-27
> >
> > **Table J. Comparison with Distributional RL and Bayesian RL on CVRP**
> >
> > |                          | Obj.↓(100) | Gap(%)↓  | Obj.↓(200) | Gap(%)↓  | Obj.↓(500) | Gap(%)↓  | Obj.↓(1000) | Gap(%)↓  |
> > | ------------------------ | ---------- | -------- | ---------- | -------- | ---------- | -------- | ----------- | -------- |
> > | Distributional RL(3000)  | 16.87      | 8.35     | 30.65      | 9.31     | 69.77      | 10.19    | 143.21      | 17.99    |
> > | Bayesian RL(3000)        | 17.02      | 9.51     | 30.93      | 10.31    | 71.77      | 13.35    | 146.15      | 21.26    |
> > | GFlowNet(3000)           | **16.53**  | **6.17** | **29.86**  | **6.49** | **68.05**  | **7.47** | **131.70**  | **9.27** |
> > | Distributional RL(5000)  | 16.78      | 7.77     | 30.47      | 8.67     | 69.29      | 9.43     | 134.29      | 11.42    |
> > | Bayesian RL(5000)        | 16.91      | 8.61     | 30.89      | 10.16    | 69.82      | 10.27    | 137.09      | 13.74    |
> > | GFlowNet(5000)           | **16.46**  | **5.72** | **29.72**  | **5.99** | **67.77**  | **7.03** | **127.40**  | **5.70** |
> > | Distributional RL(10000) | 16.73      | 7.45     | 30.38      | 8.35     | 69.24      | 9.35     | 130.74      | 8.47     |
> > | Bayesian RL(10000)       | 16.84      | 8.16     | 30.83      | 9.95     | 70.31      | 11.04    | 132.81      | 10.19    |
> > | GFlowNet(10000)          | **16.37**  | **5.14** | **29.58**  | **5.49** | **67.31**  | **6.30** | **127.26**  | **5.58** |
> >
> > [1]Distilling autoregressive models to obtain high-performance non-autoregressive solvers for vehicle routing problems with faster inference speed. AAAI.
> >
> > ------
> >
> > We sincerely thank the reviewer for the thoughtful suggestion. Your feedback has been very valuable and has helped us further improve the clarity and completeness of the paper. If you have any additional questions or suggestions, we would be more than happy to address them.

---

### Official Review · Reviewer_PDXu · 2025-10-30

**Soundness:** 2
**Presentation:** 3
**Contribution:** 2
**Rating:** 6
**Confidence:** 4

**Summary:**

This paper proposes a GFLowNet-guided 2-opt framework (AGOF) to solve routing problems, such as TSP and VRP. AGOF is an iterative framework that first takes the graph (a VRP instance) as input and outputs a scoring matrix via a GFLowNet. The scoring matrix then guides the 2-opt local search process. To further escape local optima, an exploration beyond local optima (ELO) heuristic is incorporated. Experiments on TSP and VRP instances with 100 to 1000 nodes indicate that AGOF achieves results comparable to state-of-the-art baselines.

**Strengths:**

1. AGOF is an efficient method due to its non-autoregressive design, which improves the generalization and reduces the computational cost.
2. Based on Table 1, compared to other GFLowNet-based methods, including GFACS and AGFN, AGOF achieves significant improvement.

**Weaknesses:**

1. As the authors mentioned in their paper, the goal of AGOF is to advance 2-opt-based methods. However, it is not clear that whether AGOF can outperform state-of-the-art neural network-based methods for TSP and VRP. Although the authors include several baselines beyond GFlowNet and 2-opt-based frameworks, such as LKH and POMO, incorporating more benchmark methods would be encouraged.
2. For small instances, such as TSP 100 and VRP 100, AGOF does not always outperform benchmark methods.

**Questions:**

1. Focusing on frameworks that leverage 2-opt search, such as Wu et al., DACT, and NeuOpt, it is not clear that the improvement of AGOF arises from the design of GFlowNet or the incorporation of ELO. Since all 2-opt-based methods can incorporate the usage of ELO, I would be interested in seeing the performance of 2-opt-based frameworks with ELO included.
2. In Equation (16), why not directly derive the optimal solution from the score matrix? Since the score matrix depends only on the instance graph and not on the current solution, it seems like an end-to-end approach that directly produces a solution. In this sense, the score matrix might implicitly contain enough information to infer a global solution. Could there be a way to exploit the score matrix directly to obtain the optimal solution, without the need for the time-consuming 2-opt refinement?
3. Could you conduct an additional experiment on the sensitivity of the model to the quality of the initial solution? Since a key difference between your method and previous benchmarks is that your approach does not rely on an initial solution, I am curious about how sensitive your method and the baseline methods are to the quality of the initial solution. Would providing a higher-quality initial solution lead the other methods to perform better?

---

> ### Author Response · Authors · 2025-11-27
>
> We sincerely thank you for the positive assessment of our model and for noting the advantages of AGOF’s non-autoregressive design, including its improved generalization and computational efficiency. We also appreciate the acknowledgement of AGOF’s performance improvements over previous GFlowNet-based methods such as GFACS and AGFN. Below, we provide our detailed responses to the your suggestions.
>
> ### **W1. More Baselines**
>
> Thank you for the suggestion. Following the reviewer’s advice, we have added more methods to provide a broader comparison. The additional baselines and their results are reported in Table D, and also incorporated in the Table 1 of the revised manuscript. The results show that our model outperformed these models.
>
> **Table D. Comparison with More Baseline on TSP and CVRP**
>
> |                    | Obj.↓(100) | Gap(%)↓  | Time(s) | Obj.↓(200) | Gap(%)↓  | Time(s) | Obj.↓(500) | Gap(%)↓  | Time(s) | Obj.↓(1000) | Gap(%)↓  | Time(s) |
> | ------------------ | ---------- | -------- | ------- | ---------- | -------- | ------- | ---------- | -------- | ------- | ----------- | -------- | ------- |
> | GNARKD(TSP)[1]     | 7.83       | 1.03     | 0.07    | 13.10      | 22.20    | 0.12    | 25.26      | 52.53    | 0.35    | 37.24       | 60.93    | 0.81    |
> | LCH-Regret(TSP)[2] | **7.76**   | **0.13** | 0.30    | 10.94      | 2.05     | 0.56    | 20.66      | 24.84    | 2.17    | 30.65       | 32.45    | 5.68    |
> | MDAM(TSP)[3]       | 8.38       | 7.02     | 2.79    | 14.05      | 31.06    | 5.97    | 25.31      | 52.93    | 8.79    | 39.16       | 69.23    | 19.02   |
> | AGOF(TSP)          | 7.83       | 1.03     | 0.17    | **10.90**  | **1.96** | 0.94    | **17.14**  | **3.56** | 5.15    | **24.23**   | **4.71** | 13.01   |
> | GNARKD(CVRP)       | **16.83**  | **8.09** | 0.09    | 31.72      | 13.12    | 0.18    | 76.66      | 21.07    | 0.43    | 157.25      | 30.47    | 0.92    |
> | LCH-Regret(CVRP)   | 15.72      | 9.63     | 0.35    | 30.60      | 9.13     | 0.68    | 123.06     | 94.35    | 2.18    | 372.17      | 208.78   | 36.79   |
> | AGOF(CVRP)         | 16.37      | 5.14     | 1.32    | **29.58**  | **5.49** | 1.93    | **67.31**  | **6.30** | 8.27    | **127.26**  | **5.58** | 18.27   |
> ### **W2. Performance on Small Scale Instance**
> Thank you for the thoughtful observation. While AGOF does not always surpass benchmark methods on the smallest instances, its performance at size 100 is still very encouraging. The gaps are only 0.17% on TSP100 and 1.32% on CVRP100, which remain competitive and closely aligned with strong neural baselines.
>
> More importantly, our work aims to improve the scalability and generalization of 2-opt-based approaches. From this perspective, AGOF shows a clear trend: its advantages become more evident as the problem size increases. As demonstrated in the results for sizes 200, 500, and 1000, AGOF consistently outperforms existing 2-opt-based methods. We believe this highlights the potential of AGOF to serve as a stronger foundation for learn-to-improve frameworks.
>
> ### **Q1. The Improvement of AGOF**
>
> Thank you for the question. We would like to clarify why ELO cannot be directly added to other 2-opt-based methods such as Wu et al., DACT, or NeuOpt. These methods use an autoregressive (AR) design. Their models determine the next 2-opt node pair based on the current tour and select it through the neural network. Even if a single 2-opt move does not improve the tour, the model can still choose another node pair in the next step. Because of this property, AR methods do not remain stuck at the same local solution. In other words, AR models already have a built-in ability to move away from local minima (if any), leaving no room for an additional escape mechanism such as ELO to incorporate.
>
> Our method is different. AGOF is non-autoregressive (NAR). It produces a heatmap in one forward pass and uses this heatmap to guide all 2-opt operations. When the improvement process reaches a local optimum under the current heatmap, it stops, because no better 2-opt move exists. In this case, the model cannot escape from the local minimum (if any) by itself. ELO is introduced exactly to address this situation. It provides a simple way for the NAR model to jump out of a local minimum when the refinement process can no longer make progress.
>
> To address the your concern, we also include results for AGOF without ELO and compare them with existing 2-opt-based methods. As shown in Table E, AGOF without ELO still outperformed other baseline on 200, 500, 1000. These results show that the improvement of AGOF arises from both the design of GFlowNet and the incorporation of ELO. We add these results in the Section E and Table 9 in the Appendix.

---

> > ### Author Response · Authors · 2025-11-27
> >
> > **Table E. Comparison with baselines and AGOF without ELO on TSP and CVRP**
> >
> > | Gap(%)↓    | Wu et al.(3000) | DACT(*8)(3000) | NeuOpt(3000) | AGOF(3000) | Wu et al.(5000) | DACT(*8)(5000) | NeuOpt(5000) | AGOF(5000) | Wu et al.(10000) | DACT(*8)(10000) | NeuOpt(10000) | AGOF(10000) |
> > | ---------- | --------------- | -------------- | ------------ | ---------- | --------------- | -------------- | ------------ | ---------- | ---------------- | --------------- | ------------- | ----------- |
> > | 100(TSP)   | 2.06            | 0.77           | **0.13**     | 2.97       | 1.42            | 0.52           | **0.03**     | 2.58       | 1.42             | 0.13            | **0.03**      | 2.19        |
> > | 200(TSP)   | 286.01          | 64.27          | 11.01        | **5.88**   | 277.43          | 55.78          | 9.79         | **5.41**   | 271.74           | 43.66           | 8.86          | **5.03**    |
> > | 500(TSP)   | 1004.71         | 950.15         | 700.79       | **8.82**   | 693.72          | 903.27         | 692.75       | **8.28**   | 994.14           | 837.82          | 385.98        | **7.98**    |
> > | 1000(TSP)  | 1974.46         | 1840.19        | 1252.72      | **10.03**  | 1951.25         | 1796.63        | 1242.18      | **9.85**   | 1945.89          | 1729.04         | 1231.94       | **9.29**    |
> > | 100(CVRP)  | 36.16           | 8.29           | **7.90**     | 9.18       | 30.25           | 4.54           | **1.41**     | 8.54       | 26.85            | 1.22            | **1.10**      | 5.14        |
> > | 200(CVRP)  | 47.90           | 10.27          | 60.06        | **10.08**  | 42.94           | **8.42**       | 55.42        | 9.77       | 40.51            | 6.92            | 52.92         | **5.49**    |
> > | 500(CVRP)  | -               | 14.99          | 2277.80      | **9.82**   | -               | 14.42          | 272.29       | **9.60**   | -                | 13.79           | 226.83        | **6.30**    |
> > | 1000(CVRP) | -               | -              | -            | **10.80**  | -               | -              | -            | **10.24**  | -                | -               | -             | **5.58**    |
> >
> > *"-" indicates Out-of-Memory issues for those neural baselines.*
> >
> > ### **Q2.  Derive Solution from Score Matrix**
> >
> > Thank you for the question. During training, the model learns to produce scores that are used specifically for 2-opt refinement. In other words, the learning signal is fully based on the assumption that the final solution will be decoded using 2-opt. For this reason, using 2-opt during inference is the decoding strategy that best matches the training paradigm.
> >
> > To respond to your suggestion, we tested decoding the score matrix directly using a greedy approach, and the results are reported in Table F. On TSP, greedy decoding performs slightly better than the traditional greedy benchmark, but it remains far below the performance achieved with 2-opt refinement. On CVRP, greedy decoding is even worse than the traditional greedy baseline. These results show that the score matrix is not optimized for greedy decoding, and does not support this decoding method effectively.
> >
> > In summary, the decoding strategy used at inference is implicitly learned by the model during training. Although the score matrix depends only on the instance graph and not on the current solution, it is still shaped by the 2-opt refinement process used throughout training. As a result, the model naturally expects the same decoding behavior at test time. Applying a different decoder such as greedy construction does not align with the training setup and therefore yields suboptimal performance. This explains why directly using the score matrix without 2-opt refinement is less effective than decoding in the manner consistent with the model’s training.
> >
> > **Table F. Comparison of Greedy Decode and Traditional Greedy**
> >
> > |                                    | Obj.↓(100) | Gap(%)↓   | Obj.↓(200) | Gap(%)↓   | Obj.↓(500) | Gap(%)↓   | Obj.↓(1000) | Gap(%)↓   |
> > | ---------------------------------- | ---------- | --------- | ---------- | --------- | ---------- | --------- | ----------- | --------- |
> > | Score matrix+ greedy decoder(TSP)  | **9.03**   | **16.52** | **13.27**  | **23.79** | **20.69**  | **25.02** | **28.08**   | **21.34** |
> > | Traditional Greedy(TSP)            | 9.13       | 17.81     | 12.85      | 19.87     | 20.07      | 21.27     | 28.80       | 24.46     |
> > | Score matrix+ greedy decoder(CVRP) | 23.31      | 49.71     | 38.76      | 38.23     | 81.33      | 28.44     | 158.47      | 31.48     |
> > | Traditional Greedy(CVRP)           | **20.01**  | **28.52** | **35.07**  | **25.07** | **76.80**  | **21.29** | **148.63**  | **23.31** |

---

> > > ### Author Response · Authors · 2025-11-27
> > >
> > > ### **Q3. Different Initialization Method**
> > >
> > > To study the sensitivity of different methods to the quality of the initial solution, we conducted additional experiments using greedy initialization and report the results in Table G and Table H.
> > >
> > > The results show that providing a higher-quality initial solution does not substantially change the overall conclusions. For our method, the performance with greedy initialization is almost the same across all instance sizes compared with using a random initial solution, confirming that AGOF is largely insensitive to the initial tour quality. For the other 2-opt-based baselines, greedy initialization leads to a slight improvement on the 200-, 500-, and 1000-node instances compared to initial greedy results. However, even with this advantage, their performance remains worse than AGOF. On the 100-node instances, their performance with greedy initialization is similar to that with random initialization.
> > >
> > > These results support our claim that AGOF does not rely on the initial solution and is robust to its quality, while other 2-opt-based methods benefit slightly from a better initialization but still do not reach the performance of AGOF. The results are also incorporated to revised paper in Section F and Table 10 in the Appendix.
> > >
> > > **Table G. Comparison with Greedy Initialization on TSP**
> > >
> > > |                    | Obj.↓(100) | Gap(%)↓  | Obj.↓(200) | Gap(%)↓  | Obj.↓(500) | Gap(%)↓  | Obj.↓(1000) | Gap(%)↓  |
> > > | ------------------ | ---------- | -------- | ---------- | -------- | ---------- | -------- | ----------- | -------- |
> > > | Traditional Greedy | 9.13       | 17.81    | 12.85      | 19.87    | 20.07      | 21.27    | 28.80       | 24.46    |
> > > | Wu et al.(3000)    | 7.90       | 1.94     | 12.84      | 19.78    | 20.64      | 24.71    | 28.70       | 24.03    |
> > > | DACT(*8)(3000)     | **7.80**   | **0.64** | 12.56      | 17.16    | 20.51      | 23.93    | 28.04       | 21.13    |
> > > | NeuOpt(3000)       | 7.83       | 1.03     | 11.84      | 10.45    | 20.84      | 25.92    | 28.96       | 25.15    |
> > > | AGOF(3000)         | 7.85       | 12.90    | **10.95**  | **2.15** | **17.23**  | **4.11** | **24.36**   | **5.27** |
> > > | Wu et al.(5000)    | 7.84       | 1.61     | 12.46      | 16.23    | 20.57      | 24.29    | 28.63       | 23.73    |
> > > | DACT(*8)(5000)     | **7.78**   | **0.38** | 12.32      | 14.93    | 20.34      | 22.90    | 27.83       | 20.28    |
> > > | NeuOpt(5000)       | 7.79       | 0.52     | 11.74      | 9.51     | 20.80      | 25.68    | 28.88       | 24.81    |
> > > | AGOF(5000)         | 7.82       | 0.90     | **10.93**  | **1.96** | **17.20**  | **3.93** | **24.22**   | **4.67** |
> > > | Wu et al.(10000)   | 7.84       | 1.16     | 12.27      | 14.46    | 20.52      | 23.99    | 28.61       | 23.64    |
> > > | DACT(*8)(10000)    | **7.77**   | **0.26** | 12.18      | 13.62    | 20.30      | 22.66    | 27.79       | 20.10    |
> > > | NeuOpt(10000)      | 7.78       | 0.39     | 11.71      | 9.24     | 20.78      | 25.56    | 28.84       | 24.63    |
> > > | AGOF(10000)        | 7.82       | 0.90     | **10.90**  | **1.68** | **17.15**  | **3.63** | **24.20**   | **4.58** |

---

> > > > ### Author Response · Authors · 2025-11-27
> > > >
> > > > **Table H. Comparison with Greedy Initialization on CVRP**
> > > >
> > > > |                    | Obj.↓(100) | Gap(%)↓  | Obj.↓(200) | Gap(%)↓  | Obj.↓(500) | Gap(%)↓   | Obj.↓(1000) | Gap(%)↓  |
> > > > | ------------------ | ---------- | -------- | ---------- | -------- | ---------- | --------- | ----------- | -------- |
> > > > | Traditional Greedy | 20.01      | 28.52    | 35.07      | 25.07    | 76.80      | 21.29     | 148.63      | 23.31    |
> > > > | Wu et al.(3000)    | 19.34      | 24.21    | 30.54      | 8.92     | -          | -         | -           | -        |
> > > > | DACT(*8)(3000)     | 16.68      | 7.13     | 30.66      | 9.34     | 72.73      | 14.86     | -           | -        |
> > > > | NeuOpt(3000)       | 16.60      | 6.62     | 34.36      | 22.54    | 75.02      | 18.48     | -           | -        |
> > > > | AGOF(3000)         | **16.47**  | **5.78** | **29.73**  | **6.03** | **67.85**  | **7.15**  | **130.89**  | **8.60** |
> > > > | Wu et al.(5000)    | 19.24      | 23.57    | 30.18      | 7.63     | -          | -         | -           | -        |
> > > > | DACT(*8)(5000)     | 15.82      | 1.61     | 30.42      | 8.49     | 72.17      | 13.98     | -           | -        |
> > > > | NeuOpt(5000)       | **15.77**  | **1.25** | 34.25      | 22.15    | 74.82      | 18.16     | -           | -        |
> > > > | AGOF(5000)         | 16.42      | 5.46     | **29.60**  | **5.53** | **67.43**  | **6.49**  | **127.30**  | **5.62** |
> > > > | Wu et al.(10000)   | 19.20      | 23.31    | 29.78      | 6.21     | -          | -         | -           | -        |
> > > > | DACT(*8)(10000)    | 15.75      | 1.56     | 30.03      | 7.10     | 72.72      | 14.85     | -           | -        |
> > > > | NeuOpt(10000)      | **15.75**  | **1.15** | 34.10      | 21.61    | 74.70      | 17.97     | -           | -        |
> > > > | AGOF(10000)        | 16.38      | 5.20     | **29.56**  | **5.42** | **67.26**  | **62.22** | **127.30**  | **5.60** |
> > > >
> > > > [1]Distilling autoregressive models to obtain high-performance non-autoregressive solvers for vehicle routing problems with faster inference speed. AAAI.
> > > >
> > > > [2]Learning Encodings for Constructive Neural Combinatorial Optimization Needs to Regret. AAAI.
> > > >
> > > > [3]Multi-Decoder Attention Model with Embedding Glimpse for Solving Vehicle Routing Problems. AAAI.
> > > >
> > > > ------
> > > > We sincerely appreciate your valuable suggestion and the insights it brings to strengthening our study. If you have any additional questions or further analyses you would like us to provide, we would be very happy to address them.

---

### Official Review · Reviewer_aJec · 2025-10-31

**Soundness:** 2
**Presentation:** 2
**Contribution:** 2
**Rating:** 2
**Confidence:** 4

**Summary:**

The paper addresses the problem of learning to solve vehicle routing
problems, esp. by local search in a 2-opt neighborhood. The authors
propose a method with two aspects,
- i) to predict the value of an edge (i,j) in the solution not by
  its length (distance), but by how likely it occurs in visited solution
  candidates, as estimated by a GFlowNet model,
  globally for the problem graph, independently from the current
  tour the 2-opt operator is considered to be applied to, and
- ii) to make some standard 2-opt steps to escape local minima
  (called "distance-based exploration beyond local optima").

In experiments they show that this approach performs faster than
neural solvers that estimate edge values conditonally on the
current tour, and on larger instances (e.g., CVRP500 and
CVRP1000) also much better.

**Strengths:**

- s1. approach simplifying methods from the literature: estimate edge values for 2-opt only
  globally, and not dependent on the current tour.
- s2. interesting idea: escape local minima by standard 2-opt steps instead of just adding some
  noise to the estimated egde values.
- s3. good ablation study of this escape approach vs. adding random noise (tab. 2)

**Weaknesses:**

- w1. The scope of contribution is narrow: to improve 2-opt based methods, not to
  improve methods for neural routing generally.
- w2. Comparison against OR solvers and more general neural solvers is unclear.
- w3. The description of the motivation and properties of the method is formally problematic.

more details:
w1. The scope of contribution is narrow: to improve 2-opt based methods, not to
improve methods for neural routing generally.
- Why is it interesting to improve methods specifically for 2-opt based methods?
  If these methods do not perform at the state-of-the-art, they might not be so
  interesting anymore.
- At least improvements should be clearly contextualized in the current state-of-the-art
  of OR solvers and other neural solvers (see w2 next).

w2. Comparison against OR solvers and more general neural solvers is unclear.
- LKH is known to achieve a very low gap very soon. How would it perform if you
  grant it only the time your own method uses?
- Why do you not compare against OR solver HGS?
- AGFN and POMO are much faster than your method. How do they perform if you
  let them search the same amout of time your method needs?

w3. The description of the motivation and properties of the method is formally problematic.
- You motivate your work claiming that your method estimates scores in a continuous
  way, while neural solvers conditioning on the candidate tours do not (eq. 3-9). But tours
  and graphs are discrete objects, the concept of continuity is not defined for
  them, and the formulas you show also do not describe continuity.
  It is not clear what they really mean.
- eq. 15 is unclear: the right sides does not depend on v_t. Maybe you mean
  \eta_{v_t, v_{t+1}} ?

**Questions:**

- q1. Why is it interesting to improve methods specifically for 2-opt based methods?
- q2. How to OR solvers and neural solvers such as POMO and AGFN compare to the proposed
  method if they take the same runtime?

---

> ### Author Response · Authors · 2025-11-27
>
> We sincerely thank the reviewer for recognizing these aspects of our work. In particular, we appreciate your positive feedback on our simplified global edge-value estimation strategy for 2-opt, the idea of escaping local minima through standard 2-opt steps rather than relying solely on noise injection, and the ablation study comparing this mechanism against random-noise perturbations. Your encouragement is highly appreciated and motivates us to refine the method further. Below, we provide detailed responses to the concerns you raised.
>
> ### **W1, Q1. The Scope of Contribution**
>
> In traditional operations research, improving local-improvement operators has been an important and active line of work[1][2][3]. Among these operators, 2-opt plays a particularly central role, serving as a key component in lots of strong heuristics. However, within the neural solver community, this direction has received comparatively little attention in recent years. A key reason is the bottleneck in generalization: exiting 2-opt-based learn-to-improve model often struggle to generalize to larger sizes. This limitation has discouraged further exploration of learn-based model of operators and has contributed to the community’s shift toward learn-to-construct paradigm or others instead.
>
> Our work tackles this bottleneck directly. By substantially improving the generalization ability of 2-opt within the learn-to-improve framework, our model surpasses prior 2-opt-based neural baselines and demonstrates that this research direction remains viable and valuable. We believe that addressing this core challenge can help re-invigorate interest in operator learning and pave the way for more powerful and flexible neural routing solvers in the future.
>
> ### **W2, Q2. Comparison with OR Solver and Neural Solver**
>
> **W2.1 Compare with LKH.** We include a comparison with LKH under the **same time budget**. As shown in Table A, on CVRP our method outperforms LKH on the 500- and 1000-node instances, while LKH performs better on the smaller 100- and 200-node cases. For TSP, LKH is known to produce near-optimal solutions due to its sophisticated, highly specialized design. Our model, by contrast, is only built around a simple operator (2-opt), so surpassing LKH on TSP is not an expected objective. What matters is that our method consistently improves upon all existing 2-opt–based neural baselines, which demonstrates clear progress in the learn-to-improve direction.
>
> Our contribution is not intended to replace LKH, but to enhance the generalization and effectiveness of the 2-opt operator itself. These results show that even lightweight operator-based models can remain competitive at scale when equipped with appropriate neural architectures, highlighting the value of strengthening foundational operators for neural routing research.
>
> ##### **Table A. Comparison with LKH on CVRP.**
>
> |             | Obj.(100)↓ | Gap(%)↓  | Time(s)↓ | Obj.(200)↓ | Gap(%)↓  | Time(s)↓ | Obj.(500)↓ | Gap(%)↓  | Time(s)↓ | Obj.(1000)↓ | Gap(%)↓  | Time(s)↓ |
> | ----------- | ---------- | -------- | -------- | ---------- | -------- | -------- | ---------- | -------- | -------- | ----------- | -------- | -------- |
> | LKH         | **15.85**  | **1.80** | 0.50     | **29.65**  | **5.74** | 0.67     | 69.60      | 10.27    | 2.75     | 134.74      | 11.79    | 5.88     |
> | AGOF(3000)  | 16.53      | 6.17     | 0.43     | 29.86      | 6.49     | 0.70     | **68.05**  | **7.47** | 2.44     | **131.70**  | **9.27** | 5.49     |
> | LKH         | **15.80**  | **0.32** | 0.75     | **29.04**  | **3.57** | 1.22     | 68.72      | 8.53     | 4.32     | 131.52      | 9.12     | 9.75     |
> | AGOF(5000)  | 16.46      | 5.72     | 0.75     | 29.72      | 5.99     | 1.17     | **67.77**  | **7.03** | 4.31     | **127.40**  | **5.70** | 9.56     |
> | LKH         | **15.78**  | **0.19** | 1.40     | **28.75**  | **2.53** | 1.99     | 67.39      | 6.43     | 8.52     | 127.61      | 5.87     | 18.67    |
> | AGOF(10000) | 16.37      | 5.14     | 1.32     | 29.58      | 5.49     | 1.93     | **67.31**  | **6.30** | 8.27     | **127.26**  | **5.58** | 18.27    |
>
> **W2.2 Comparison with HGS.**
> As noted earlier, our aim is to improve the generalization of 2-opt–based neural models, not to outperform carefully engineered solvers such as HGS, which incorporates a highly sophisticated combination of genetic algorithm, population management strategies, adaptive penalty mechanisms, and local search optimization. We include LKH and HGS only as strong reference points to contextualize the performance of neural baselines, including ours.
>
> Following the reviewer’s suggestion, we additionally incorporate HGS results in Table B. Since HGS is designed specifically for CVRP, we report comparisons on CVRP instances accordingly.

---

> > ### Author Response · Authors · 2025-11-27
> >
> > **Table B. Comparison with HGS.**
> >
> > |            | Obj.(100)↓ | Gap(%)↓ | Time(s)↓ | Obj.(200)↓ | Gap(%)↓ | Time(s)↓ | Obj.(500)↓ | Gap(%)↓ | Time(s)↓ | Obj.(1000)↓ | Gap(%)↓ | Time(s)↓ |
> > | ---------- | ---------- | ------- | -------- | ---------- | ------- | -------- | ---------- | ------- | -------- | ----------- | ------- | -------- |
> > | HGS        | 15.50      | -       | 16.81    | 27.83      | -       | 87.79    | 62.26      | -       | 705.80   | 118.05      | -       | 4369.41  |
> > | AGOF(1000) | 16.37      | 5.61    | 1.32     | 29.58      | 6.29    | 1.93     | 67.31      | 8.11    | 8.27     | 127.26      | 7.80    | 18.27    |
> >
> > **Summary.** LKH and HGS integrate decades of expert design and sophisticated mechanisms, whereas our method is intentionally grounded in a simple operator-based framework. It is therefore natural that our approach does not exceed such mature OR solvers in general. Nevertheless, by substantially improving the generalization and performance of 2-opt, our work provides a meaningful foundational advancement. More importantly, the enhanced operator can potentially be incorporated into higher-level frameworks, including those used in LKH and HGS, which may offer a complementary and extensible contribution.
> >
> > **W2.3 Comparison with AGFN and POMO.** AGFN and POMO belong to the learn-to-construct family of methods. They directly build complete routes using the neural network, without any improvement or refinement steps. Since they do not perform iterative optimization, their runtime is naturally much shorter than that of learn-to-improve approaches, which deliberately allocate time to iterative refinement.
> >
> > This design, however, may also limit their potential: once a solution is constructed, these models cannot make further progress even if more computation time is available. In contrast, learn-to-improve methods, including ours, start from random initial solutions and continually refine them, allowing solution quality to improve with additional computation. These differences reflect two distinct design philosophies: learn-to-construct methods prioritize speed, while learn-to-improve methods pursue higher solution quality by making full use of extra computation time.
> >
> > We have added the corresponding explanations and experimental results in the revised manuscript, specifically in the updated Section D and Table 7-8 in the Appendix.
> >
> > ### **W3.  Description of the Motivation and Properties of the Method**
> >
> > To clarify our use of the term "continuous scoring", we do not refer to continuity of the tour space, which is clearly discrete. Instead, our claim refers to a broader notion of continuity: when the input graph undergoes a small modification, the model’s action(for AR) or heatmap(for NAR), scoring function and the resulting trajectory reward change only slightly and in a stable manner. In this context, continuity characterizes how stable the model’s behavior remains under small perturbations to the input graph. Following this definition, the equation in the paper is  to show that the reward of AR trajectories can vary significantly under such perturbations, whereas the NAR formulation exhibits much smaller changes. We will add this definition to the revised manuscript to make the intended meaning clearer.
> >
> > In addition, we provide an empirical study showing that small changes in the input graph lead to much smaller variations in the resulting reward for our model compared with other AR models. Specifically, we add a single node to the datasets of size 100, 200, 500, and 1000 and measure the resulting change in the final reward. The results, presented in Table C, show that our model exhibits significantly smaller reward fluctuations than other 2-opt based AR methods. These results have been incorporated into the revised paper in Section 4.4 and Table 5 in the Appendix.

---

> > > ### Author Response · Authors · 2025-11-27
> > >
> > > **Table C. Comparison of Reward Sensitivity to Graph Perturbation**
> > >
> > > |                 | ΔObj.↓(100)(TSP) | ΔObj.↓(200)(TSP) | ΔObj.↓(500)(TSP) | ΔObj.↓(1000)(TSP) | ΔObj.↓(100)(CVRP) | ΔObj.↓(200)(CVRP) | ΔObj.↓(500)(CVRP) | ΔObj.↓(1000)(CVRP) |
> > > | --------------- | ---------------- | ---------------- | ---------------- | ----------------- | ----------------- | ----------------- | ----------------- | ------------------ |
> > > | Wu et al.(3000) | 0.306            | 0.452            | 0.573            | 0.648             | 0.194             | 0.297             | -                 | -                  |
> > > | DACT(*8)(3000)  | 0.095            | 0.496            | 0.528            | 0.615             | 0.131             | 0.135             | 0.166             | -                  |
> > > | NeuOpt(3000)    | 0.038            | 0.359            | 0.376            | 0.391             | 0.176             | 0.508             | 0.870             | -                  |
> > > | AGOF(3000)      | **0.028**        | **0.017**        | **0.011**        | **0.015**         | **0.128**         | **0.122**         | **0.120**         | **0.126**          |
> > >
> > > *"-" indicates Out-of-Memory issues for those neural baselines.*
> > >
> > > **Eq. 15.** Thank you for pointing this out. You are correct, the right-hand side of Eq. 15 should depend on $v_t$. The correct form is indeed $\eta_{v_t, v_{t+1}}$. We have updated the equation accordingly in the revised version.
> > >
> > > [1]A hybrid adaptive iterated local search with diversification control to the capacitated vehicle routing problem[J]. European Journal of Operational Research.
> > >
> > > [2]PILS: Exploring high-order neighborhoods by pattern mining and injection[J]. Pattern Recognition.
> > >
> > > [3]A novel feature-based approach to characterize algorithm performance for the traveling salesperson problem[J]. Annals of Mathematics and Artificial Intelligence.
> > >
> > > ------
> > > We sincerely appreciate the reviewer’s comments and suggestions, which have helped us improve the clarity of the manuscript. If the reviewer has any further questions or additional points that require clarification, we would be very glad to address them.

---

> > > > ### Comment · Reviewer_aJec · 2025-11-27
> > > >
> > > > Thanks to the authors for their answers to my questions.
> > > >
> > > > w1. narrow contribution focused on 2-opt.
> > > > - I agree that researching 2-opt is interesting. I just would have liked to see further
> > > >   state-of-the-art solvers compared, both local search based and construction
> > > >   based.
> > > >
> > > > w2. comparison with OR and neural solvers.
> > > > - The comparison with LKH at same runtime budget is really interesting, because it shows,
> > > >   that for large instances you have a case.
> > > > - The comparison with HGS suffers from the same issue as LKH originally: you do not
> > > >   compare at the same runtime budget.
> > > > - Comparison with constructive solvers is problematic: you always can combine any constructive
> > > >   solver with a local search, initializing the local search with the solution found by the
> > > >   constructive method.
> > > >
> > > > w3. motivation and description of the method
> > > > - I understand what you want to say. But you cannot phrase it as "continuity".
> > > >   The formal part clearly needs thorough rewriting, in my opinion.
> > > >
> > > > In my opinion the paper could benefit a lot from improving on the last two
> > > > points, and thus stay with my score.

---

> > > > > ### Author Response · Authors · 2025-12-03
> > > > >
> > > > > We sincerely thank the reviewer for the thoughtful and constructive feedback. Your comments on the comparison baselines, runtime fairness, and the clarity of the method description have been extremely valuable for improving our work. We have carefully revised the manuscript to address these points and strengthened both the experimental evaluation and the formal presentation of the method. Our detailed responses are provided below.
> > > > > ### **W1 Comparison with Gurobi**
> > > > > Based on your comment, we also found it valuable to compare our method with OR solvers under an equal time budget, since traditional learning-based approaches typically find it challenging to outperform strong OR solvers. To this end, we additionally provide results for the exact solver Gurobi[1], which is one of the most established OR solvers.
> > > > >
> > > > > Due to the extremely large scale of the 10,000-instance dataset for size 100, we were unable to run Gurobi on all cases within a reasonable time frame. Therefore, we report Gurobi’s performance on 128 instances for sizes 200 and 500 with a 10-minute time limit each. For the 1000-node setting, even with a 60-minute time limit, Gurobi was unable to find feasible solutions on these instances. As shown in Table K, AGOF surpasses Gurobi on the 200- and 500-node instances while also being substantially more efficient.
> > > > >
> > > > > **Table K. Comparison with Gurobi**
> > > > > |             | Obj.(200)↓ | Gap(%)↓  | Time(s)↓ | Obj.(500)↓ | Gap(%)↓  | Time(s)↓ | Obj.(1000)↓ | Gap(%)↓  | Time(s)↓ |
> > > > > | ----------- | ---------- | -------- | -------- | ---------- | -------- | -------- | ----------- | -------- | -------- |
> > > > > | Gurobi      | 29.91      |6.68 | 600      | 73.18      |15.52 | 600      | - |  - | 3600 |
> > > > > | AGOF(10000) | **29.58**  | **5.49** | 1.93     | **67.31**  | **6.30** | 8.27     | **127.26**  | **5.58** | 18.27    |
> > > > >
> > > > > ### **W2.1 Comparison with LKH**
> > > > > We thank you for the suggestion. It is particularly encouraging to see that our learning-enhanced 2-opt operator, which remains structurally simple, is able to outperform LKH on large instances, even though LKH relies on a substantially more complex algorithmic design.
> > > > >
> > > > > ### **W2.2 Comparison with HGS**
> > > > > We have provided the HGS results under the same (or close) time budget for your reference. However, as you are also aware, it is inherently challenging for a single 2-opt operator to surpass HGS, given its extremely sophisticated design and the decades of engineering effort behind it. The corresponding results are reported in Table B.1.
> > > > >
> > > > > **Table B.1 Comparison with HGS under the Same Time Budget**
> > > > > |             | Obj.(100)↓ | Gap(%)↓ | Time(s)↓ | Obj.(200)↓ | Gap(%)↓ | Time(s)↓ | Obj.(500)↓ | Gap(%)↓  | Time(s)↓ | Obj.(1000)↓ | Gap(%)↓  | Time(s)↓ |
> > > > > | ----------- | ---------- | ------- | -------- | ---------- | ------- | -------- | ---------- | -------- | -------- | ----------- | -------- | -------- |
> > > > > | HGS         | 15.80      |1.48 | 0.46     | 28.70      | 2.35| 0.86     | 65.12      | 2.84 | 2.74     | 123.07      | 2.11 | 6.30     |
> > > > > | AGOF(3000)  | 16.53      | 6.17    | 0.43     | 29.86      | 6.49    | 0.70     | 68.05  | 7.47 | 2.44     | 131.70  | 9.27 | 5.49     |
> > > > > | HGS         | 15.76      | 1.22 | 0.83     | 28.68      | 2.28  | 1.26     | 65.06      | 2.75 | 4.39     | 122.97      | 2.02 | 10.67    |
> > > > > | AGOF(5000)  | 16.46      | 5.72    | 0.75     | 29.72      | 5.99    | 1.17     | 67.77  | 7.03 | 4.31     | 127.40  | 5.70 | 9.56     |
> > > > > | HGS         | 15.70      |0.83| 1.45     | 28.59      | 1.96  | 2.06     | 64.87      |2.45 | 8.89     | 122.80      | 1.88  | 18.45    |
> > > > > | AGOF(10000) | 16.37      | 5.14    | 1.32     | 29.58      | 5.49    | 1.93     | 67.31  | 6.30 | 8.27     | 127.26  | 5.58 | 18.27    |

---

> > > > > > ### Author Response · Authors · 2025-12-03
> > > > > >
> > > > > > ### **W2.3 Comparison with AGFN with Local Search**
> > > > > > Following your suggestion, we additionally report the performance of the GFlowNet-based constructive method AGFN [2] combined with local search, evaluated under the same (or close) time budget as AGOF. As shown in Table L, our method consistently outperforms AGFN across all instance sizes.
> > > > > >
> > > > > > **Table L. Comparison with AGFN Enhanced by Local Search under the Same Time Budget**
> > > > > > |                        | Obj.(100)↓ | Gap(%)↓ | Time(s)↓ | Obj.(200)↓ | Gap(%)↓ | Time(s)↓ | Obj.(500)↓ | Gap(%)↓  | Time(s)↓ | Obj.(1000)↓ | Gap(%)↓  | Time(s)↓ |
> > > > > > | ---------------------- | ---------- | ------- | -------- | ---------- | ------- | -------- | ---------- | -------- | -------- | ----------- | -------- | -------- |
> > > > > > | AGFN with local search | 16.97      | 9.01    | 0.46     | 30.59      | 9.10    | 0.76     | 69.21      | 9.30     | 2.57     | 132.90      | 10.26    | 5.62     |
> > > > > > | AGOF(3000)             | **16.53**    | **6.17**  | 0.43   |**29.86**    | **6.49**  | 0.70   | **68.05**  | **7.47** | 2.44     | **131.70**  | **9.27** | 5.49     |
> > > > > > | AGFN with local search | 16.89      | 8.48    | 0.79     | 30.38      | 8.35    | 1.26     | 68.95      | 8.89     | 4.39     | 130.45      | 8.23     | 9.68     |
> > > > > > | AGOF(5000)             | **16.46**    | **5.72**  | 0.75   | **29.72**    | **5.99**  | 1.17  | **67.77**  | **7.03** | 4.31     | **127.40**  | **5.70** | 9.56     |
> > > > > > | AGFN with local search | 16.78      | 7.78    | 1.38     | 30.33      | 8.17    | 2.04     | 68.59      | 8.28     | 8.32     | 129.72      | 7.62     | 18.36    |
> > > > > > | AGOF(10000)            | **16.37**    | **5.14**  | 1.32  | **29.58**    | **5.49**  | 1.93   | **67.31**  | **6.30** | 8.27     | **127.26**  | **5.58** | 18.27    |
> > > > > >
> > > > > >
> > > > > >
> > > > > > ### **W3 Concern about 'continuity'**
> > > > > > Thank you very much for this helpful comment. We fully understand your concern, and we agree that the original phrasing using “continuity” was not precise. Following your suggestion, we have thoroughly revised the formal description, with the main corrections made in Section 3.1. Specifically, We replaced “continuity” with the more appropriate term "robustness" and provide detailed explanation of "robustness", which better reflects our intended meaning and makes the paper more clear. We think these updates improve the clarity and convey our intended message more effectively.
> > > > > >
> > > > > > [1]Gurobi Optimizer Reference Manual. Gurobi Optimization, LLC.
> > > > > >
> > > > > > [2]Adversarial Generative Flow Network for Solving Vehicle Routing Problems. ICLR.
> > > > > >
> > > > > > ---
> > > > > > We sincerely appreciate the reviewer’s insightful suggestions. Your comments helped us recognize that our 2-opt–based model, although fundamentally simple, can surpass extremely sophisticated OR solvers on certain instance scales, that was difficult to imagine for previous learning-based approaches. We believe this finding carries meaningful implications for the broader VRP community and highlights the value of revisiting simple operators through a learning-enhanced perspective.

---

### Meta-Review · Area_Chair_4wMx · 2026-01-01

**Summary:**

This paper proposes a non-autoregressive algorithm for solving Vehicle Routing Problems named AGOF by leveraging the 2-Opt heuristic. It can effectively resolve the generalization problem in learning-based 2-opt methods. Experimental results on TSP and VRP instances with 100 to 1000 nodes show that AGOF achieves results comparable to state-of-the-art baselines.

Clearly, this work is not bad. It makes contributions to advance the specific line of 2-Opt-based neural methods. However, all reviewers indicated the major issue, i.e., incomprehensive comparison with OR solvers and SOTA neural solvers. Some reviewers also indicated that the performance advantage is not significant on small instances.

The authors made great efforts to address the above issues by adding additional experiments. Some of the concerns can be addressed to some extent. But I think a major revision is still required, to make the contribution more convincing (especially more comprehensive comparison with OR solvers and SOTA neural solvers). Hope the comments are helpful for the authors to improve the paper and make a stronger submission in the future.

**Reviewer Concerns:**

Reviewer aJec has major concerns: 1) narrow contribution as only 2-opt based methods are improved; 2) unclear comparison against OR solvers and more general neural solvers; 3) problematic description of the motivation and properties of the method.

The authors gave detailed responses, added additional experimental results, and made clarification about 'continuity'. I think some of the reviewer’s concerns can be addressed. But the reviewer thought a major revision is required to improve the paper: especially more comprehensive comparison with OR solvers and SOTA neural solvers.

Reviewer PDXu has major concerns: 1) comparison with more benchmark methods; 2) the advantage is not clear for small instances.

The authors gave detailed responses, and added additional experimental results. I think the concerns can be addressed to some extent.

Reviewer bCtj has major concerns: 1) no comparison with solvers like Gurobi and other NAR methods applicable to VRP problems; 2) the advantage is not significant on small and medium-scale VRPs (<500).

The authors gave detailed responses, and added additional experimental results. I think the concerns can be addressed to some extent.

Reviewer pYHa has major concerns: 1) no comparion with the well-known general neural VRP solver; 2) fairness on the runtime comparison.

The authors gave detailed responses, and added additional experimental results. I think the concerns can be addressed to some extent.

**Reviewer Scores:**

I think some of the concerns of Reviewer aJec can be addressed. However, she/he kept the score after reading the response.

I think the concerns of Reviewer PDXu can be addressed to some extent. The reviewer may keep the positive score.

I think the concerns of Reviewer bCtj can be addressed to some extent. The reviewer may increase the score slightly.

I think the concerns of Reviewer pYHa can be addressed to some extent. The reviewer may keep the positive score.

---

### Decision · Program_Chairs · 2026-01-26

Reject